

# How important is biomass burning in Canada to mercury contamination?

Annemarie Fraser[1], Ashu Dastoor[2], and Andrei Ryjkov[2]

[1]Air Quality Research Division, Environment and Climate Change Canada, 335 River Road, Ottawa, Canada

[2]Air Quality Research Division, Environment and Climate Change Canada, 2121 Trans-Canada Highway, Dorval, Quebec, Canada H9P 1J3

*Correspondence to*: Ashu Dastoor (ashu.dastoor@canada.ca)

**Abstract.** Wildfire frequency has increased in past four decades in Canada, and is expected to increase in future as a result of climate change (Wotton et. al. 2010). Mercury (Hg) emissions from biomass burning are known to be significant; however, the

impact of biomass burning on air concentration and deposition fluxes in Canada has not been previously quantified. We use estimates of burned biomass from FINN (Fire Inventory from NCAR) and vegetation-specific Emission Factors (EFs) of mercury to investigate the spatiotemporal variability of Hg emissions in Canada. We use Environment and Climate Change Canada's GEM-MACH-Hg (Global Environmental Multi-scale, Modelling Air quality and Chemistry model, mercury version) to quantify the impact of biomass burning in Canada on spatiotemporal variability of air concentrations and deposition fluxes of

mercury in Canada. We use North American gaseous elemental mercury (GEM) observations (2010-2015), GEM-MACH-Hg, and an inversion technique to optimize the emission factors for GEM for five vegetation types represented in North American fires to constrain the biomass burning impacts of mercury. We use three biomass burning Hg emissions scenarios in Canada to conduct three sets of model simulations for 2010-2015: two scenarios where Hg is emitted only as GEM using literature or optimized EFs, and a third scenario where Hg is emitted as GEM using literature EFs and particle bound mercury (PBM) emitted

using the average GEM/PBM ratio from lab measurements. The three biomass burning emission scenarios represent the range of possible values for the impacts of Hg emissions from biomass burning in Canada on Hg concentration and deposition.

We find total biomass burning Hg emissions to be highly variable from year to year, and estimate average 2010-2015 total atmospheric biomass burning emissions of Hg in Canada to be between 6-14 t during the biomass burning season (i.e., from May to September), which is 3 - 7 times the mercury emission from anthropogenic sources in Canada for this period. On average,

65% of the emissions occur in the provinces west of Ontario. We find that while emissions from biomass burning have a small impact on surface air concentrations of GEM averaged over individual provinces/territories, the impact at individual sites can be as high as 95% during burning events. We estimate average annual mercury deposition from biomass burning in Canada to be between 0.3 – 2.8 t, compared to 0.14 t of mercury deposition from anthropogenic sources during the biomass burning season in Canada. Compared to the biomass burning emissions, the relative impact of fires on mercury deposition is shifted eastward, with

on average 54% percent of the deposition occurring in provinces west of Ontario. While the relative contribution of Canadian biomass burning to the total mercury deposition over each province/territory is no more than 9% between 2010-2015, the local contribution in some locations (including areas downwind of biomass burning) can be as high as 80% (e.g. northwest of Great Slave Lake in 2014) from May-September. We find that northern Alberta and Saskatchewan, central British Columbia, and the area around Great Slave Lake in the Northwest Territories are at greater risk of mercury contamination from biomass burning.

GEM is considered to be the dominant mercury species emitted from biomass burning; however, there remains an uncertainty in the speciation of mercury released from biomass burning. We find that the impact of biomass burning emissions on mercury deposition is significantly affected by the uncertainty in speciation of emitted mercury because PBM is more readily deposited




closer to the emission sources than GEM; an addition of ~18% percent of mercury emission from biomass burning in the form of PBM in the model increases the 6 year average deposition by ~ 4 times.

## 1 Introduction

Mercury is a trace metal found throughout the environment. It is of concern because, once released to the atmosphere, it can be deposited and transformed to methylmercury, a potential neurotoxin in animals and humans (AMAP/UNEP, 2013). Mercury is emitted from natural sources such as volcanoes and weathering of mercury-containing rocks, anthropogenic sources such as the burning of coal and metals smelting, and through the reemission of mercury historically deposited from anthropogenic and natural sources onto soils, surface water, and vegetation (AMAP/UNEP, 2013). Atmospheric mercury exists in three forms: gaseous elemental mercury (GEM), gaseous oxidized mercury (GOM), and particle bound mercury (PBM). Total atmospheric mercury (TAM) is the total of all three of these species, total gaseous mercury (TGM) is the total of GEM and GOM, and total oxidized mercury (TOM) is the total of GOM and PBM (CMSA, 2016). Total global Hg emissions from current anthropogenic sources were estimated at about 2000 t/year in 2010, with the overall shares of GEM, GOM, and PBM emissions equal to 81, 15, and 4%, respectively (AMAP/UNEP 2013). Coal combustion, and artisanal and small-scale gold mining were the major anthropogenic sources of mercury emissions to the atmosphere accounting for about 25 and 37%, respectively (AMAP/UNEP 2013). Geographically, East and Southeast Asia account for approximately 40 % of the global anthropogenic emissions of mercury. In comparison with major mercury-emitting countries, Canadian anthropogenic emissions are relatively small (e.g., 5.3 t/year in 2010) and represent less than 0.3 % of global anthropogenic emissions. Global Hg emissions other than current anthropogenic sources from soils and oceans (mainly emitted as GEM) are estimated to be 5207 t/year, which represent nearly 70% of the global mercury emission budget with oceans and biomass burning contributing to the most emissions at 36 % and 9 %, respectively (Pacyna et al. 2016; De Simone et al., 2015).

Biomass burning is one of the pathways previously emitted mercury can be reemitted (Friedli et al., 2001; De Simone et al., 2015). Atmospheric mercury is transported to vegetation and soils directly via both dry and wet deposition (Webster et al., 2016; Jiskra et al., 2015). Some mercury in the soil is taken up by vegetation, with the amount dependent on the vegetation type (Fay and Gustin, 2007). During fires, mercury in vegetation and soils is released to the atmosphere, primarily as GEM, and possibly with a sizeable proportion (up to 30%) as PBM (Obrist et al., 2008). The amount of mercury released during burning depends on the severity of the fire and the amount of mercury in the biomass and soils being burned. The severity of the fire in turn depends on the available fuel, atmospheric conditions, and forest structure (Webster et al., 2016).

Early attempts to quantify the large-scale emissions from biomass burning were made from government estimates of fuel consumption and mercury emission factors (e.g. Friedli et al., 2001; Cinnirella and Pirrone, 2006; Weiss-Penzias et al., 2007, Chen et al., 2013). These methods are limited by the location of the ground-based fuel consumption measurements, which cannot capture fires where there were no observations. With the development of fuel consumption inventories based on satellite measurements, global and regional estimates of mercury release from biomass burning became possible, which cover a wider geographical area than ground-based measurements but are limited to cloud-free scenes (e.g. Wiedinmyer and Friedli, 2007; Cinnirella et al., 2008; Friedli et al., 2009; Huang et al., 2011). Global estimates of mercury released from biomass burning vary by an order of magnitude from 100 to 1000 tonne/year (CMSA, 2016), with an estimated 22 tonne/year released from boreal regions of North America (Friedli et al., 2009).

Once released, mercury is primarily removed from the atmosphere through dry and wet deposition. GOM and PBM are soluble, and so readily removed by precipitation and dry deposition. GEM is relatively insoluble, and is removed slowly both





through direct dry deposition mainly to vegetation, and via conversion to TOM in the atmosphere, which is then removed via dry and wet deposition (AMAP/UNEP, 2013). GOM and PBM have relatively short lifetimes in the atmosphere of up to two weeks, thus deposit on a regional scale; whereas GEM has a longer lifetime of up to a year, thus transports and deposits on a global scale (AMAP/UNEP, 2013). Wet deposition is easily observable by analysing the mercury content of collected precipitation, but dry

deposition is more complicated to measure (CMSA, 2016). As a result, most dry and total deposition estimates come from modelling studies. Total deposition in Canada, calculated from a modelling study using GEM-MACH-Hg, is on the order of 120 t/year (CMSA, 2016). Approximately 40% of the mercury deposited was from current global anthropogenic emissions of which over 95% came from foreign sources, and approximately 60% from other global terrestrial (approximately 35%) and oceanic (approximately 25%) emissions (CMSA, 2016). East Asia and the United States were the predominant areas contributing to

Canada's mercury burden.

There have been few studies of deposition from biomass burning emissions: De Simone et al. (2015) compared the resulting deposition from three biomass burning inventories, finding that over 75% of mercury emitted from biomass burning is deposited to the oceans. De Simone et al. (2017) investigated the effects of including PBM emitted from biomass burning, and found that this fraction is reduced to 62-71%, indicating that more mercury is deposited closer to the fires, on land, when PBM is

emitted in the model. The frequency of large fires, and thus area burned, in Canada has increased in the last four decades of the 20[th] century (Fauria and Johnson 2008). Changes in forest fires may be the greatest early impact of climate change on forests. Increases in fire occurrences in Canada have been attributed in part to global warming (e.g., Gillett et al. 2004), and are projected to increase in future (Wotton et. al. 2010). Since the Canadian landscape contains significant boreal forests which are estimated to have the largest rate of Hg emissions during fires of all biomes (Friedli et al., 2009), mercury emissions from biomass burning

source in Canada are expected to be significant; however, the impact of biomass burning on air concentration and deposition fluxes in Canada has not been previously quantified. The objective of this study is to use the 3-D GEM-MACH-Hg atmospheric model and biomass burning emissions of Hg based on FINN (Fire INventory from NCAR, Wiedinmyer et al., 2006, 2011) to estimate the impact of biomass burning in Canada on mercury concentrations and deposition in Canada. The spatiotemporal variability in biomass burning mercury emissions and its impacts are examined over a 6 years period, i.e. 2010-2015.

Canadian oil sands operations in Northern Alberta are reported to release industrial mercury emissions to the air along with other anthropogenic emissions such as $NO_X$ and $SO_2$ (NPRI, htpps://www.ec.gc.ca/inrp-npri/), and there is a concern that ecosystem-wide mercury concentrations are on the rise in the region, affecting the food-web, as a result of industrial development in the area (Timoney and Lee, 2009). However, Northern Alberta and Saskatchewan are also known to be frequently impacted by wildfires. Parsons et al. (2013) analysed measured TGM concentrations at the monitoring station Fort

McMurray (Patricia McInnes air quality monitoring station) in northern Alberta, a station approximately 30 km south of major Canadian oil sands developments, from October 2010 to May 2013. The authors found TGM surface air concentrations close to background level and no significant correlation between ambient concentrations of TGM and other anthropogenic pollutants at Fort McMurray. The authors noted that the highest TGM concentrations observed at the site are a result of forest fire smoke. In contrast, Kirk et al. (2014) found higher than background concentrations of mercury deposited in snow in the Canadian oil sands

region, likely as a result of oxidized Hg emissions from the oil sands operations. The investigation of biomass burning impacts in our study forms the first part of a comprehensive study to investigate and isolate major sources of Hg contamination in and around the Canadian oil sands.

In our study, we use FINN (Wiedinmyer et al., 2006, 2011) estimates of burned biomass together with vegetation-specific emission factors (EFs) to generate emissions estimates of mercury released from biomass burning in North America. The

EFs are the ratio of mercury emitted to biomass consumed (Webster et al., 2016, Wiedinmyer and Friedli, 2007). These factors



have been derived for North America from laboratory- and field-studies of representative vegetation types, and are not well-characterized (Wiedinmyer and Friedli, 2007; CMSA, 2016). We use ground-based observations of TGM and GEM from across North America, the GEM-MACH-Hg atmospheric model, and a synthesis inversion technique to optimize the EFs of mercury for vegetation types represented in North America to constrain the FINN derived mercury emissions from biomass burning in North

America. We then use the prior and optimized biomass burning emissions of Hg in the model to characterize the resultant mercury burden in the air and deposited to the surface in each of the provinces and territories in Canada. In Sect. 2 and Sect. 3 we describe the data and models, respectively. In Sect. 4 we describe our method for identifying fire episodes in the model and observations. In Sect. 5 we find the optimized emission factors, and in Sect. 6 we compare the results from the model runs with the prior and posterior biomass burning emissions of mercury. In Sect. 7 we examine the mercury burden from biomass burning

in Canada. Finally, we conclude the paper in Sect. 8.

## 2 Data

We use TGM or GEM data from 16 Canadian and 26 American observation stations across Canada and the United States. The American sites are operated as part of the Atmospheric Mercury Network (AMNet, Steffen et al., 2012). The Canadian sites are part of the Environment and Climate Change Canada's (ECCC's) Canadian Atmospheric Mercury Measurement Network

(CAMNet), the Canadian Air and Precipitation Monitoring Network (CAPMoN), the Joint Oil Sands Monitoring Program (JOSM), the Northern Contaminants Program (NCP), the Canadian Clean Air Regulatory Agenda (CARA), and two special studies (Cole et al., 2014). All the sites studied here use a Tekran 2537 instrument to measure TGM or GEM concentrations (Kos et al., 2013). Figure 1 shows the location of all the Canadian stations and the American stations where forest fire plumes were observed. The filled symbols indicate stations where biomass burning was detected during our study period. The names,

locations, and network information for these stations is given in Table 1. Despite some observations being of TGM, we interpret all measurements as being GEM for comparison with the model, as the difference between the two is normally no more than a few percent (Slemr et al., 2015).

## 3 Models

### 3.1 GEM-MACH-Hg

GEM-MACH-Hg is the mercury version of the ECCC's current operational air-quality forecast model GEM-MACH (Global Environmental Multi-scale, Modelling Air quality and CHemistry model) (Makar et al., 2015a,b; Gong et al., 2015; Makar et al., 2017; Moran et al, 2010). GEM-MACH-Hg is an online model: the meteorology is simulated in-step with the chemistry, and includes representation of physicochemical processes of mercury based on the ECCC's previous mercury model, GRAHM (Global/Regional Atmospheric Heavy Metals model, Dastoor and Larocque, 2004; Dastoor et al., 2008, 2015; Durnford et al.,

2010, 2012; Kos et al., 2013). We use the model with a horizontal resolution of 10 km in a nested grid over North America. The grid extends from Northern Mexico to near the tip of Ellesmere Island in Northern Canada, and from Eastern Iceland to the Bering Strait separating Alaska and Russia. Boundary conditions for the nested model are driven by a global 1°x1° run of the same model. GEM (Hg(0)) is oxidized to GOM and PBM in the atmosphere by OH and bromine (Sommar et al., 2001; Donohoue et al., 2006; Dibble et al., 2012; Goodside et al., 2004). No aqueous-phase reduction reactions are included; however,

the OH reaction rate constant is scaled down by a coefficient of 0.34 to take into account dissociation reactions (Tossell, 2003; Goodsite et al, 2004). OH fields are taken from MOZART (Model for OZone and Related chemical Tracers, Emmons et al.,





2010) while BrO is derived from 2007-2009 satellite observations of BrO vertical columns. The associated Br concentration is calculated by assuming photochemical steady state between Br and BrO (Platt and Janssen, 1995). GEM-MACH includes a full tropospheric chemistry reaction mechanism, and particle chemistry and microphysics (Makar et al., 2015a,b); the OH concentrations are used for oxidation of GEM and are derived from this oxidation mechanism.

5        Dry deposition of GEM, GOM and PBM is included based on the resistance approach (Zhang, 2001; Zhang et al., 2003). Wet deposition of Hg is included by partitioning GEM and GOM between liquid cloud droplets and air using a temperature-dependent Henry's Law constant, and PBM is scavenged by cloud droplets and snow crystals (Durnford et al. 2012). A dynamic multilayer snowpack/meltwater mercury parameterization allowing the representation of deposition and re-emission of mercury is used (Durnford et al. 2010). Oceanic evasion of GEM is activated if there is open water and the temperature at the

air-sea interface is -4℃ or greater and depends on the distribution of primary production and atmospheric deposition (Dastoor and Durnford, 2014).

       Emissions from natural sources and re-emissions of previously deposited mercury are based on global budgets (Gbor et al., 2007; Shetty et al., 2008; Mason, 2009). Natural emissions are spatially distributed according to the natural enrichments of mercury. Land re-emissions are spatially distributed according to the historic deposition and land-use type and depend on solar

radiation and the leaf area index.

       For anthropogenic mercury emissions, a hybrid set of inventories was used for Canada: emissions from major point sources are based on the 2013 National Pollutant Release Inventory (NPRI, htpps://www.ec.gc.ca/inrp-npri/), emissions from the Upstream Oil and Gas sector (oil exploration and production) are based on the Clearstone 2011 inventory (Clearstone Engineering Ltd., 2014), and transportation and area source emissions are based on the 2010 Air Pollutant Emission Inventory

(APEI, http://www.ec.gc.ca/Pollution/). For mercury emissions in the United States, the 2011 National Emissions Inventory (NEI, htpps://www.epa.gov/air-emissions-inventories/2011-national-emissions-inventory-nei-data) was used. In both countries, SMOKE emission processing system was used to provide the diurnal variation of the emissions. Total anthropogenic emissions of Hg used in the model in Canada and the United States were 5.3 t/year and 47 t/year, respectively. The model simulation on global scale was conducted with anthropogenic Hg emissions for 2010 (1880 t/year) obtained from AMAP (AMAP/UNEP,

2013). Details on the emissions of the anthropogenic species aside from mercury may be found in Zhang et al. (this issue).

       Biomass burning emissions are derived from the FINN fire emissions product (Wiedinmyer et al., 2006, 2011). We take the burned biomass derived by FINN and multiply it by a vegetation-type specific emission factor to derive biomass burning emissions of mercury as GEM. We use the FINN vegetation types (VTs), which are derived from the MODIS Land Cover Type (Friedl et al., 2010) and grouped into seven generic VTs, and the EFs from Wiedinmyer and Friedli (2007), which are given in

Table 2. The seven VTs are: grassland, woody savanna, tropical forest, temperate forest, boreal forest, temperate needleleaf forest, and crops. In our base model we input all of the emissions as GEM.

### 3.2 Inverse model

Synthesis, or Bayesian inversions, are a well-established top-down technique for optimizing emissions of atmospheric species such as carbon monoxide (e.g. Palmer et al., 2003), methane (e.g. Wang et al., 2004), and carbon dioxide (e.g. Fraser et al.,

2014), and have also recently been used in mercury studies. Song et al. (2015) used a Bayesian inversion to optimize Asian anthropogenic mercury emissions. Other inverse methods have been used to study mercury emissions. de Foy et al. (2012, 2014) used an inverse modelling technique to optimize mercury emissions upwind of Milwaukee, Wisconsin. Pan et al. (2007) used 4DVar to optimize mercury emissions in China. Roustan and Bocquet (2006) used an adjoint technique to optimize mercury emissions in Europe.





Considering significant uncertainties in biomass burning emissions of mercury, we constrain the FINN-derived biomass burning Hg emissions in North America by optimizing the vegetation-specific emission factors used to construct the emissions fields used in the model in order to make the modelled Hg concentrations better match the GEM observations. To achieve this, we use an inverse model to find the maximum a posteriori solution (MAP, Rodgers, 2000). We assume that the emission factors

are constant in time, that the burned biomass estimates from FINN are correct, and that changes in the emission factors are linearly related to changes in the modelled mercury concentrations. We use the ground-based observations of mercury described in Sect. 2 and the GEM-MACH-Hg model described in Sect. 3.1.

We are seeking the solution to a linear problem with forward model:

$$\mathbf{y} = \mathbf{F}(\mathbf{x}) + \boldsymbol{\epsilon} = \mathbf{K}\mathbf{x} + \boldsymbol{\epsilon}, \tag{1}$$

where **y** is a vector containing the daily-averaged GEM observations with error $\boldsymbol{\epsilon}$, and **F(x)** is the forward model that describes how **x**, the emission factors, are related to the observations. **K** is the linearization of the forward model and is described in more detail below. The MAP solution to this problem is given by the posterior emission factors ($\hat{\mathbf{x}}$) and posterior error covariances ($\hat{\mathbf{S}}$):

$$\hat{\mathbf{x}} = \mathbf{x}_a + \left(\mathbf{K}^T \mathbf{S}_\epsilon^{-1} \mathbf{K} + \mathbf{S}_a^{-1}\right)^{-1} \mathbf{K}^T \mathbf{S}_\epsilon^{-1} (\mathbf{y} - \mathbf{K}\mathbf{x}_a) \tag{2}$$

and:

$$\hat{\mathbf{S}} = \left(\mathbf{K}^T \mathbf{S}_\epsilon^{-1} \mathbf{K} + \mathbf{S}_a^{-1}\right)^{-1}, \tag{3}$$

where $\mathbf{x}_a$ is a vector containing the prior emission factors, $\mathbf{S}_\epsilon$ is the observation error covariance matrix, $\mathbf{S}_a$ is the prior EF covariance matrix, and **K** is the Jacobian matrix. In practice we are assuming that the system is linear for small changes, and we are solving for changes to the EFs to match the difference between the modelled and observed concentrations, so we take the last

bracketed term in Eq. (2) to be $(\mathbf{y} - \mathbf{y}_m)$, where $\mathbf{y}_m$ is the vector of daily-averaged modelled concentrations, sampled at the same time and location as the observations.

The prior emission factors, $\mathbf{x}_a$, are taken from Wiedinmyer and Friedli (2007), as described in Sect. 3.1. This vector has a length of six, for the six vegetation types that we attempt to optimize. We do not optimize VT3 (tropical forests) because there are very few detected fires that contain burning from tropical forests. We construct the prior error covariance matrix, $\mathbf{S}_a$, as a

diagonal matrix (dimension 6×6) with the elements the square of the error in the prior EFs, which we assume to be 100%. We assume that the errors in the EFs are uncorrelated, and so the off-diagonal elements of the matrix are all zero.

We construct the observation vector, **y**, from GEM observations during fire events. From the time series of the model ($\mathbf{y}_m$) and observations ($\mathbf{y}$) during the fire, we define a time period where the fire is detected. We take the mercury observation of the fire to be the daily-averaged observations over that time period. We construct the model vector, $\mathbf{y}_m$, in the same way. These

vectors have a length of 268, the number of days with fires that we have identified in our six-year dataset (see Sect. 4).

We construct the observation error covariance matrix, $\mathbf{S}_\epsilon$, as a diagonal matrix (dimension 268×268) with the elements the square of the measurement error. We take the measurement error to be the sum in quadrature of the instrumental error and the mean station model-mismatch error (Wang et al., 2004). The model-mismatch error attempts to account for model transport error, and sub-grid scale variations in Hg concentrations. We take the instrumental error to be a conservative 10% (e.g. Cole et

al., 2013; Song et al., 2015).

The Jacobian matrix, **K**, represents the sensitivity of the modelled concentrations to a change in the emission factors:

$$K_{i,j} = \frac{\partial y_i}{\partial EF_j}. \tag{4}$$

We construct **K** by running a series of sensitivity runs of the model. For each year we run the model with a perturbation to an individual emission factor of $10 \times 10^{-6}$ g Hg (kg fuel burned)$^{-1}$. We repeat this for all six emission factors, resulting in 36





sensitivity runs (6 years × 6 vegetation types). We then sample these model simulations at the time and location of the observations. The dimension of this matrix is 268 × 6.

## 4 Fire Events Identification

To identify fire plumes in the modeled and observed time series of GEM air concentrations at the observation sites, we run the
model once with the complete global Hg emissions as described in Sect. 2, and once with all of the emissions except biomass burning Hg emissions in North America (i.e., the "no fire" run). The difference of these model runs gives us the GEM concentration as a result of only the biomass burning emissions in North America. We sample this difference in GEM concentration at the time and location of the observations. GEM peaks in these station time series indicate times when the model predicts a fire plume at one of the stations. We compare the model simulated fire only GEM concentration to the observations
with the mean of the "no fire" simulation GEM concentration subtracted. We define a fire event as any time the fire only model and observations have peaks that are within a day of one another, with a maximum value of the modelled GEM concentration greater than twice the standard deviation of the "no fire" modelled GEM concentration for that year and station. The GEM peaks that are predicted by the model but not found in the observed data are attributed to model transport error or errors in the FINN burned biomass inventory and not errors in the emission factors, and these fire events are not considered in our analysis. Also,
GEM peaks that are in the observations but not in the model output we assume to be from sources other than biomass burning. Using the model we follow the plume back in time to identify the source region of the fire. In total, we find 30 fire plumes in our six year dataset totalling 268 burning days, which are shown in Figure 2. In the case of fire plumes observed at one station that are a result of fires in two different locations, we separate the fire plume into two or more events, based on the approximate time the second plume arrives at the station. Three of the fire plumes are sub-divided to make 35 fire events. This sub-division does
not factor into our analysis, which uses the daily mean of observed and modelled GEM concentrations, and is done only for informational purposes.

Fire plumes observed at sites near Fort McMurray (FTM, AMS, LWC) are mostly from fires located near Great Slave Lake to the north in the Northwest Territories, or near Lake Athabasca in Alberta and Saskatchewan. Fire plumes observed at sites in British Columbia (MBL, SAT, WSL) and at GEN are from fires located in the British Columbia forests. Fire plumes observed at
AK03 and FL96 originate from fires located within Alaska and Florida, respectively. The fire plume observed at VT99 originates from a fire in Northwestern Quebec. The fire plume observed at EGB in 2011 originates in Northwestern Ontario, while the fire plumes observed there in 2014 were transported from fires in the Northwest Territories. Examples of these fire plumes are shown in Figure 3. While we identify the primary location of the fire by following the plume backwards, the contribution of each vegetation type to each fire is not constrained to the location of these fires.

30        Figure 4 shows the fractional GEM concentration contributions from each vegetation type to each of the 35 fire events. The product of the Jacobian matrix, **K**, and the prior emission factors, **x**, gives the contribution of each vegetation type to the total concentration during the fires. Dividing this by the total concentration gives the fractional contribution. All of the vegetation types are represented by the observed fires; however most fires are dominated by VT5, the boreal forest. This is to be expected from the location of the fires in northern and western Canada, where boreal forests are the dominant vegetation type.





## 5 Optimized emission factors

The optimized emission factors and associated errors are given in Table 2. Since we mainly use observed concentrations of GEM for the optimization procedure, the optimized EFs are considered to represent biomass burning emissions of GEM. We also provide the percentage error reduction, γ, defined as:

$$\gamma = \left[1 - \frac{\epsilon}{\epsilon_0}\right] \times 100\%, \tag{5}$$

where $\epsilon$ is the posterior error and $\epsilon_0$ is the prior error (*e.g.* Fraser et al., 2013). Larger values of γ indicate that more information has been extracted from the observations. The emission factor for VT5, boreal forests, has the largest error reduction. This is to be expected given the fractional contributions shown in Figure 4. VT2, woody savanna, has a small error reduction, indicating that the posterior does not greatly improve on the prior value. This can be further explored by studying the averaging kernel, **A**, given by (Rodgers, 2000):

$$A = \left(K^T S_\epsilon^{-1} K + S_a^{-1}\right)^{-1} K^T S_\epsilon^{-1} K. \tag{6}$$

In an ideal case, the rows of **A** would all peak at one, with no contribution from the other elements (i.e. a unit matrix). The area of the averaging kernel gives a measure of the fraction of the retrieval that comes from the observations. Elements with an averaging kernel near one are therefore desirable (Rodgers, 2000). The rows of **A** and their area are shown in Figure 5. The VT5 averaging kernel is near ideal – with a peak at almost one and an area of 1.01. VT6, temperate evergreen forest, and VT7, crops, have well-defined peaks with not much influence from the other vegetation types, and areas of 0.98 and 0.88, respectively, meaning that most of the information in the posterior estimate of these emission factor comes from the observations. The other vegetation types are not as well resolved. VT4, temperate forest, has an area of 0.52, but with some influence coming from VT2. VT1, grasslands, shows a strong peak of about 0.6, and an area of 0.73, but has large influence from VT2. VT2, woody savannas, has a small area of 0.12 with a very small peak. The influence of VT2 on the other vegetation types is an indication that there is very little independent observation of VT2 in the observed fire plumes: fires that contain this vegetation type also contain one or more of the others. Because of this analysis, and the small values of the error reduction for this vegetation type, we perform the inversion for a second time, without optimizing for VT2. The results of this inversion are also shown in Table 2, and we use these values as the posterior EFs.

By any measure, VT5, the boreal forest, is the EF best constrained by our method. The value is reduced from 315 to $(140 \pm 27) \times 10^{-6}$ g Hg (kg fuel burned)$^{-1}$. The prior value is taken from Wiedinmyer and Friedli (2007), and is an average of seven studies where the emission factors were measured in experimental campaigns. Six of the seven studies range in mean value between 60 and $207 \times 10^{-6}$ g Hg (kg fuel burned)$^{-1}$, with one study reporting $1476 \times 10^{-6}$ g Hg (kg fuel burned)$^{-1}$ (Turetsky et al, 2006). Removing the smallest and largest values of these seven, and taking the average of the remaining five studies, yields an average of $134 \times 10^{-6}$ g Hg (kg fuel burned)$^{-1}$, in agreement with the $(140 \pm 27) \times 10^{-6}$ g Hg (kg fuel burned)$^{-1}$ found in our work. The outlying value from Turetsky et al. (2006) is itself an average of values ranging between 535 - $2417 \times 10^{-6}$ g Hg (kg fuel burned)$^{-1}$, and was measured in fires over boreal peatlands in western Canada. Their own range for fires not over peatlands is $90 - 297 \times 10^{-6}$ g Hg (kg fuel burned)$^{-1}$, an average of $193 \times 10^{-6}$ g Hg (kg fuel burned)$^{-1}$. Our assigned vegetation types, taken from the FINN inventory, do not take peatland into account, though vast areas of north and western Canada are peatland (Tarnocai et al., 2011).

VT6, temperate needleleaf forest, is the next best constrained EF in our study (error reduction 74%), and the posterior value increases to $(315 \pm 62)$ from $239 \times 10^{-6}$ g Hg (kg fuel burned)$^{-1}$. This prior value is calculated as a mean of eight studies, with values ranging from 80 to $654 \times 10^{-6}$ g Hg (kg fuel burned)$^{-1}$. Limiting the studies to the four that took place in the western United States, where the majority of the fires in VT6 occur, yields a mean of $356 \times 10^{-6}$ g Hg (kg fuel burned)$^{-1}$. The prior for





VT4, temperate forest, is taken from the same eight studies. The posterior value for VT4 is $(254 \pm 181) \times 10^{-6}$ g Hg (kg fuel burned)$^{-1}$, with an error reduction of 29%. This is not much changed from the prior, and the error reduction shows that most of the information in the posterior value is coming from the prior information.

The error reductions for VT7, crops, and VT1, grassland, are 53% and 38%, respectively, and the posterior values are $(215 \pm 129)$ and $(213 \pm 170) \times 10^{-6}$ g Hg (kg fuel burned)$^{-1}$, reduced from a common prior of $274 \times 10^{-6}$ g Hg (kg fuel burned)$^{-1}$. Only two studies are averaged to generate the prior, with values of 38 and $510 \times 10^{-6}$ g Hg (kg fuel burned)$^{-1}$.

## 6 Biomass burning emissions

We generate optimized mercury emissions for North America using the posterior emission factors in Table 2. We conduct model simulations for the six-year study period using optimized biomass Hg emissions as GEM in North America and other global Hg emissions. We refer to this model simulation as the posterior model. We then sample the posterior model GEM at the time and location of the GEM observations. Figure 2 shows the comparison of daily averaged GEM concentrations from observations and from model runs with the prior and posterior biomass burning mercury emissions for all fire plumes.

Table 3 provides Pearson correlation coefficient (r), root mean square error (RMSE), unbiased root mean square error (URMSE), and normalised standard deviation (NSD) values between model simulated and observed surface air concentrations of GEM using all observation sites calculated for all individual years studied and collectively for all years using prior and posterior biomass burning emission factors of Hg. For the collective values, the agreement between the observations and the model is slightly improved by using the optimized emission factors. However, for individual years, r is better using prior EFs for 3 years and improves for the other 3 years using posterior EFs. Examining the individual fire events in Figure 2 reveals that the posterior EFs generally improve the agreement between the model and observations in 2010, have a neutral effect in 2011, 2012 and 2013, and generally degrade the agreement in 2014 and 2015. This is an indication of a break down in one or more of our initial assumptions: the FINN calculation of burned biomass has uncertainties in magnitude or in location, the emission factors are not constant in space-time but are functions of fire type and other factors such as atmospheric deposition, or the six vegetation types do not accurately represent the variation in mercury emissions by species. Fires observed in 2010 are mainly located in British Columbia (BC), while those in 2014/15 are mainly located around Great Slave Lake. Peat is much more prevalent in the Northwest Territories than in BC (Tarnocai et al., 2011), the discrepancy in improvement between the years is perhaps an indication that peatland should be considered in defining the vegetation types; this is currently difficult due to sparseness of measurements of Hg from biomass burning plumes. Given small and inhomogeneous differences in the model-measurement agreement between the prior and posterior biomass burning emission scenario model runs, we use both prior and posterior EFs for biomass burning emissions of Hg to provide a range of possible values for the impact of biomass burning on mercury concentrations and deposition, acknowledging that depending on the individual fire, one inventory or the other is more representative of the true emissions.

Figure 6 shows the total GEM emissions from biomass burning, both absolute and relative to the total Canadian biomass burning emissions of GEM, over the burning season (i.e., May – September) for each province and territory for 2010-2015 and for the mean of the six year period. Prior biomass burning emissions of mercury over the burning season range from 10.8 t in 2011 to 15.6 t in 2014 (with 6 year average of 12.2 t), while posterior emissions range from 5.0 t in 2011 to 7.5 t in 2014 (with 6 year average of 5.8 t). There are large inter-annual variations in the total emissions and the relative contribution from the provinces/territories, reflecting the variation in the location and duration in forest fires over this time period. For example, the biomass burning emissions of Hg are significantly larger in Ontario, Quebec and North-West Territories in 2011, 2013 and 2014,



respectively, compared to the emissions from these regions in other years. This can also be seen in Figure 7, which shows maps of the prior biomass burning emissions of mercury for each of the six years of the study. While the absolute value of the emissions is reduced in the posterior emissions by half, the relative contribution to total biomass burning emissions in Canada is mostly unchanged. This is due to the fact that while the EFs are changed by different factors by the inversion, the emissions in

Canada are dominated by emissions from fires in the boreal forest, which is roughly halved.

Fire in-plume mercury measurements generally found mercury emissions to be mostly in the form of GEM (Friedli et al. 2003; Sigler et al. 2003); for example, Friedli et al. (2003) measured PBM and GOM fractions to be ~ 0.8% and 0.0%, respectively. However, lab-based measurements of mercury species released from biomass burning show highly variable PBM concentrations ranging from not detectable to up to 30% of the total emitted mercury (Obrist et al., 2008; Zhang et al., 2013).

The speciation is likely variable within the fire, and depends on the vegetation burned, the intensity of the fire, and the moisture content of the fuel (Obrist et al., 2008). De Simone et al. (2017) were able to better reproduce observed PBM concentrations at two remote sites by including biomass burning emissions of PBM in North America in a global modeling study. We generated an additional set of biomass burning emissions of mercury in North America by adding mercury emissions of PBM to the prior biomass Hg burning emission scenario such that the GEM emissions are at the same value as in prior GEM emission scenario

and the GEM to PBM emissions ratio is 8.5:1.5 which is an average of observed ratios found in Obrist et al. (2008), and also used in De Simone et al. (2017). We chose to keep the total GEM biomass burning emissions as in the prior emission scenario since the literature values of Hg EFs are thought to be for emissions of GEM. Prior GEM with PBM biomass burning emissions of mercury in Canada over the burning season range from 12.7 t in 2011 to 18.4 t in 2014 with an average of 14.3 t over 2010-2015.

Figure 6 also shows the total mercury emissions from anthropogenic sources in Canada during the biomass burning season. The total anthropogenic emissions over the same five months are roughly 2 t; 3-7 times less the emissions from biomass burning. The relative contribution from the provinces/territories is different as well, with roughly half the anthropogenic emissions coming from provinces east of Manitoba, half coming from provinces west of Ontario, and virtually no emissions from the northern territories. Biomass burning emissions, averaged over the six study years, are dominated by emissions from the

western provinces (~65%), with the remaining 35% split between the eastern provinces and northern territories.

Figure 8 shows the map of the monthly averaged anthropogenic emissions during the burning season. The spatial distributions of biomass burning and anthropogenic emissions are quite different. The bulk of the anthropogenic emissions occur in population centres, while the emissions from biomass burning are generally located away from cities.

## 7 Biomass burning impacts on mercury burden in Canada

We perform three sets of model simulations by using biomass burning Hg emissions with prior GEM EFs (the "prior GEM" run), posterior GEM EFs (the "posterior GEM" run), and prior GEM EFs with PBM (the "prior with PBM" run) in Canada along with other global emissions of mercury for 2010-2015. Using these three model experiments along with a model simulation without biomass burning Hg emissions in Canada, we evaluate the contributions of biomass burning Hg emissions in Canada to the surface air concentration and deposition of mercury for 2010-2015 during burning season. PBM is more readily deposited by

wet and dry deposition, closer to the sources; whereas GEM is transported and slowly oxidized in air to TOM and deposited on a regional scale through the direct dry deposition of GEM and through the wet and dry deposition of TOM (CMSA, 2016). The model simulation with additional PBM biomass burning emissions allows us to investigate the effects of PBM released from biomass burning on the mercury burden. We perform this model simulation only with the prior biomass burning emissions of



mercury to estimate the upper limit of possible impacts of biomass burning emissions. Considering significant uncertainties in EFs of mercury and the speciation of mercury emissions from biomass burning, we provide the range of possible values of the impact of mercury emissions from biomass burning in Canada using the three biomass burning emission scenarios.

We also compare the contributions from the three biomass burning emissions scenarios on Hg burden in Canada to the contributions from Canadian anthropogenic Hg emissions and total Hg mercury emissions. We examine the contributions at local, provincial/territorial and national level, and in the case of concentrations in the air, at the individual observation sites from 2010-2015.

### 7.1 Impact on air concentrations

The daily percentage contribution of the Canadian biomass burning emissions to the total average surface air GEM concentration in each province/territory for each burning season from 2010-2015 is shown in Figure 9. Because the GEM emissions are the same, there is no significant difference between the prior run with and without the contribution from PBM. While the mean contribution to the surface concentration is small in both inventories (<2.5%), there are spikes in this contribution in all provinces. The highest such spike is 31%, seen in the prior emissions in Manitoba in 2015. Emissions from biomass burning in Manitoba were not significant that year (see Figure 6), but emissions in Saskatchewan, directly upwind, were the largest in that province over the study period. Peaks in the contribution from biomass burning occur during burning events (e.g. Northwest Territories in 2014), but the largest peaks are from the transport of burning from upwind (e.g. Alberta in 2010). These contributions are averaged over the province/territory, so areas of the regions that are not influenced by burning contribute to lowering the concentration. This figure also shows the percentage contribution of Canadian anthropogenic emissions to the total concentrations in each province/territory. These contributions are more constant, reflecting the more consistent emissions. Using the prior biomass burning emissions, the overall contribution from biomass burning is larger than the contribution from anthropogenic emissions in all regions. Using the posterior emissions, this is true in the provinces west of Quebec and the Northwest Territories, while in the remaining provinces and territories the contribution from anthropogenic sources is equal or larger.

Figure 10 shows the contribution of the Canadian biomass burning and anthropogenic emissions to the total average surface TOM (GOM + PBM) concentration in each province/territory for each burning season. Here, as expected, the contribution from the prior with PBM run is much larger than the other two emissions scenarios. For these scenarios with no PBM emissions from biomass burning, TOM is produced only by the slow oxidation of emitted GEM from biomass burning, which is a very small contribution (<0.3%). The average contributions from the prior with PBM run are not large (<5.5%), but spikes as large as 55% (again in Manitoba in 2015) are possible. The contribution from anthropogenic emissions to TOM air concentration is always larger than that from biomass burning in the prior and posterior runs with no PBM. This is because anthropogenic emissions have contribution from TOM while these biomass burning runs do not. For most provinces/territories the average anthropogenic contribution to TOM concentrations is larger than the contribution to TOM concentrations from the prior run with PBM, the exceptions being British Columbia, Alberta, and Ontario. In all regions, the peaks in TOM concentrations contributions in the prior run with PBM can be several times larger than those from anthropogenic emissions.

Figure 11 shows the daily percent contribution of the Canadian biomass burning emissions to the total surface air GEM concentration at the Canadian mercury observation sites listed in Table 1 and shown in Figure 1. Here the percentage contribution is as high as 95%, which occurs at sites AMS and LWC (both are in the oil sands region) during a local fire event in 2011. The largest peaks in the contribution from biomass burning occur during times when there are fires near the observation site (e.g AMS, FTM, and LWC in 2011). Sites where there were no fire plumes observed over the study period (e.g. HFX, KEJ)


have smaller contributions from biomass burning, but there is still some influence due to long-range transport of mercury from the fires. This figure also shows the percent contribution of the Canadian anthropogenic emissions. As for the biomass burning contribution, the percentage contributions from anthropogenic emissions are larger at individual stations than averaged over the provinces/territories, reaching as high as 15% at BRL, FFT, and WBZ. Which source is responsible for the largest share of emissions is site-dependant, and reflects the location of the observation station with respect to the source of emissions. Figure 12 shows the same as Figure 11 but the total surface air TOM concentrations. Again, the average contribution from the prior and posterior runs with no PBM is quite small (<1%). The average contribution from the prior with PBM run is also small (<4%), but has peaks of up to 95% at AMS and LWC during burning events. The contribution from anthropogenic emissions varies significantly, from less than a few percent at remote sites such as ELA and LFL, but as high as 100% at sites close to the emissions, such as FTM and LWC. Day-to-day variations at one site can be as large as 100%. At all sites except for LFL, the average contribution from anthropogenic emissions is larger than the contribution from biomass burning.

**7.2 Impact on deposition**

The total deposited mercury from the Canadian biomass burning source during the burning season (i.e., May-September) for 2010-2015, as well as the mean for all years, for each province/territory from prior GEM (a), posterior GEM (b), and prior GEM with PBM (c) emission scenarios model simulations are shown in Figure 13. Total mercury deposition from biomass burning source in Canada ranges from 0.5 in 2011 to 0.8 t in 2014 (with an average 0.6 t) for the prior GEM scenario, 0.2 in 2011 to 0.4 t in 2014 (with an average 0.3 t) for the posterior GEM scenario, and 2.4 in 2011 to 3.8 t in 2014 (with an average 2.8 t) with prior GEM with PBM biomass burning emission scenario compared to 0.14 t of deposition from Canadian anthropogenic source during the burning season. Averaged over 6 years, Canadian biomass burning contributions to Hg deposition in Canada are 4.5, 2, and 20 times higher than Canadian anthropogenic contributions using prior GEM, posterior GEM and prior GEM with PBM scenarios, respectively, during the biomass burning season. In line with the emissions from biomass burning in Figure 6, there is inter-annual variation in the deposited mercury from biomass burning in terms of both the absolute and relative contribution, reflecting the variation in the location and duration of the forest fires; however the variation is not as large as that of the emissions, because a significant portion of the emitted mercury is transported out of the region (~ 95% for prior and posterior with GEM scenarios and ~80% for prior with PBM scenario). As expected, the mercury deposition from the posterior with GEM biomass burning emissions is roughly half of the prior with GEM biomass burning emissions; however, the deposition contribution from prior GEM with PBM scenario is 4.4 times higher than prior GEM scenario, a result of PBM being more readily deposited than GEM, and so more mercury is deposited closer to the biomass burning source. This indicates that the speciation of the mercury emitted from biomass burning has a significant impact on the amount of mercury deposited, which is consistent with De Simone et al., 2017. Comparing the regional contributions of the emissions in Figure 6 and the deposition in Figure 13, the relative contributions to the deposition are different. The provinces west of Ontario are responsible for, on average, 67% of the emissions, but only 54% of the deposition. The provinces east of Manitoba are responsible for only 14% of the emissions but 28% of the deposition. The emissions in the northern territories release 9% of the emissions, but account for 18% of the deposition. This reflects the general circulation of the atmosphere. This behaviour is also seen in comparing the spatial distribution of the biomass mercury emissions in Figure 7

Figure 7 and the spatial distribution of biomass burning deposition from the run with PBM in Figure 14. Deposition mainly occurs downwind of the sources. Figure 13 (d) shows 6 year average Canadian biomass burning and anthropogenic deposition contributions to the total Hg deposition in each province/territory for the prior with PBM run during the biomass burning period. The contribution of the biomass burning deposition from the run with PBM by province/territory to the total deposition from all



sources in that province/territory during the biomass burning season is no more than 9% (not shown), which occurs in the Northwest Territories in 2014, a year with maximum forest fires in this region. In comparison, the deposition contribution from Canadian anthropogenic source to total Hg deposition in a province is highest in Alberta at 0.3%.

Figure 15 shows the spatial distribution of the average monthly percentage contribution of the deposition from Canadian biomass

burning emissions ("prior with PBM" model simulation) to the total deposition from all sources in that grid box during the biomass burning season. While the relative contribution of Canadian biomass burning to total deposition averaged over each province/territory and 2010-2015 is no more than 7%, (Figure 13 (d)), the local contribution in some locations can be as high as 80% (e.g. northwest of Great Slave Lake in 2014) from May-September. While the location of these biomass burning deposition "hotspots" changes from year-to-year depending on the location of biomass burning, during the six-year study period, regions

that have a large percentage of the local deposition coming from biomass burning include northern Alberta and Saskatchewan, central British Columbia, and the area around Great Slave Lake in the Northwest Territories.

We can compare the modelled mercury deposition from Canadian biomass burning source in Figure 14 to the deposition from global biomass burning source in Canada found in De Simone et al. (2015). This study investigated the impact of mercury emissions from biomass burning for 2006-2010 from three inventories (i.e., FINN, GFAS, and GFED) where Hg emissions were

assumed to be emitted only as GEM. De Simone et al. (2017) modelled mercury deposition from biomass burning for 2013 by partitioning the mercury emissions between GEM and PBM as 85:15. In both studies mercury deposition was estimated for Canada in the range of 0.5-2 µg/m$^2$/year. Our simulated deposition from biomass burning in Canada agrees well with this estimate, but varies in a wider range, i.e., 0.05-2.5 µg/m$^2$/year, which is most likely due to differences in model resolutions - 2.5°×2.5° in De Simone et al. (2017) versus 10x10 km in GEM-MACH-Hg. Spatial distributions of Hg deposition by biomass

burning are also similar between De Simone et al. (2017) and our study - e.g., both studies simulate higher deposition in 2013 in the Northwest Territories and Norther Quebec.

Figure 16 shows the spatial distribution of the Hg deposition from Canadian anthropogenic sources in 2012 (other years are similar). In contrast to biomass burning contribution, about 60% Hg deposition from the Canadian anthropogenic source occurs in provinces east of Manitoba. Deposition primarily occurs near the source regions in Southern Ontario and

Quebec, around Vancouver in southeastern BC, in Central Alberta, and near the Saskatchewan-US border compared to the deposition from biomass burning which occurs downwind of the forest fires. While the deposition from anthropogenic sources does not vary significantly from year to year, the deposition from biomass burning is highly variable, depending on the location of the fires.

## 8 Discussion and conclusions

We have used the GEM-MACH-Hg model, the network of North American GEM/TGM observation sites, and an inversion technique to optimize emission factors for five vegetation types represented in North American fires to estimate the biomass burning emissions of mercury in Canada and its impact on mercury contamination in Canada from 2010-2015. The observed data provides the most information about the EF for boreal forests, which is the most constrained by our technique. We find this EF is reduced to $(140 \pm 27) \times 10^{-6}$ kg Hg/kg biomass burned from $315 \times 10^{-6}$ kg Hg/kg biomass burned, which is within

the range of values found in the literature. However, we find that optimized EFs fail to improve the simulations of episodic high surface air GEM concentrations during fire events for all years at all observation sites studied. We attribute this discrepancy to the uncertainties in FINN calculated burned biomass amounts and location, and the assumption that the biomass burning EFs of Hg can be represented only by the six vegetation types used in this study. It is also likely that EFs are functions of fire type and



other factors such as atmospheric deposition. For example, peat is much more prevalent in the Northwest Territories than in BC (Tarnocai et al., 2011), the discrepancy in improvement between the years is perhaps an indication that peatland should be considered in defining the vegetation types; this is currently difficult due to sparseness of measurements of Hg in biomass burning plumes. Acknowledging that depending on the individual fire, one inventory or the other is more representative of the

true emissions, we conduct model simulations using both prior and posterior EFs for biomass burning emissions of Hg. Mercury released from vegetation and soils to the atmosphere during fires is primarily found to be in the form of GEM (Friedli et al. 2003; Sigler et al. 2003); however, lab-based measurements of Hg emissions from biomass burning from various types of biomass fuels are shown to emit variable amounts of PBM in addition to GEM (up to 30%; Obrist et al., 2008; Zhang et al., 2013). In order to estimate the impact of PBM emissions from biomass burning, we conduct an additional model simulation by adding PBM

emissions to prior GEM emission scenario with a GEM/PBM ratio of 8.5:1.5. We consider the model results from the three biomass burning scenarios to represent the range of possible values for the impacts on mercury concentration and deposition from biomass burning mercury emissions in Canada.

We find that biomass burning is a significant source of mercury emissions in Canada. The total mercury emitted from biomass burning during the burning season of May-September over the six years of our study ranges from 5.0 to 7.5 t for the

posterior GEM emission scenario, 10.8 to 15.6 t for the prior GEM emission scenario, and 12.7 to 18.4 t for the prior GEM with PBM emission scenario. Averaged over 2010-2015, biomass burning Hg emissions in Canada are 3-7 times higher than the mercury emitted from anthropogenic sources in Canada during the burning season depending on the biomass burning emission scenario. The spatial distribution of biomass burning emissions is variable from year to year, but is always very different from the spatial distribution of the anthropogenic emissions, which are mostly located near population centres. On average, 65% of

the biomass burning emissions occur in the provinces west of Ontario, but this ranges from 30-90% over our six-year study. Saskatchewan emits the most mercury from biomass burning averaged over the six years, but again this is highly variable, with Alberta, Saskatchewan, Quebec, and the Northwest Territories having the largest emissions in individual years.

Using the prior, posterior and additional PBM biomass burning emission scenarios, we investigate the impacts of Canadian biomass burning on the air concentration and deposition in Canada. We find that, averaged over the burning season,

GEM emissions from biomass burning contribute no more than 2% to the mean surface air GEM concentration in any province/territory, but this percentage can be as high as 30% downwind of burning events. At individual measurement sites, the contribution to GEM can be as high as 95% during burning events. When PBM is directly emitted from fires, in the prior GEM with PBM run, the contribution of Canadian biomass burning emissions to the TOM concentrations is no more than 5.5% in any province/territory, but reaches as high as 95% at individual measurement sites.

Deposition from biomass burning in Canada is also significant, ranging from 0.2 to 0.4 t for the posterior GEM, 0.5 to 0.8 t for the prior GEM, and 2.4 to 3.8 t with the inclusion of PBM to prior GEM biomass burning emission scenario; averaged over 6 years, this is 2, 4.5, and 20 times higher than the contribution from Canadian anthropogenic source during the biomass burning season, respectively. Compared to the emissions, the relative contribution of the provinces/territories is shifted eastward, or downwind, of the emissions, with 54% of the deposition occurring in provinces west of Ontario. The spatial

distribution of the deposition from biomass burning and anthropogenic emissions is very different, with anthropogenic emissions depositing near population centres, while biomass burning emissions deposit in remote locations. Areas downwind of biomass burning can have up to 80% of the local deposition come from biomass burning emissions during May-September. We find that northern Alberta and Saskatchewan, central British Columbia, and the area around Great Slave Lake in the Northwest Territories are at greater risk of mercury contamination from biomass burning.





The impact of inclusion of PBM emission on Hg deposition from biomass burning source is noteworthy; 18% increase in the amount of mercury emitted from the fires in the form of PBM increases the average 2010-2015 yearly deposition from biomass burning by 4.4 times. This is because PBM is more readily deposited regionally compared to GEM. There remains significant uncertainty into the magnitude of the emissions released from biomass burning. The amount of biomass burned,

which underpins the emissions estimates in this work, has uncertainties related to the detection of small fires, land cover, satellite overpass timing, and estimated burned area (Wiedinmyer et al., 2011). Our synthesis inversion study could be improved upon by implementing a more detailed optimization scheme, for example by considering more vegetation/land-use types such as peatland into consideration when assigning vegetation types and by accounting for spatial distribution of atmospheric deposition. Comprehensive measurements of mercury species in biomass burning emission plumes for different land-use types, and a

suitable network of air concentration measurements of mercury including speciation would help in constraining the estimates of the Hg emissions from biomass burning and the resulting deposition.

**Acknowledgements**

This study was supported by ECCC and the Joint Oil Sands Monitoring program (JOSM) . We acknowledge the ECCC's Canadian National Atmospheric Chemistry (NAtChem) Toxics Database, the contributing networks and programs including the

Canadian Atmospheric Mercury Measurement Network (CAMNet), the Canadian Air and Precipitation Monitoring Network (CAPMoN), JOSM, the Northern Contaminants Program (NCP), and the Canadian Clean Air Regulatory Agenda (CARA), and the contributing scientists for the provision of Canadian mercury measurement data used in this study. We acknowledge the Atmospheric Mercury Network (AMNet) and the contributing scientists for the provision of mercury measurement data for the US sites used in this study. We acknowledge Canadian National Pollutant Release Inventory (NPRI), the Clearstone 2011

inventory, and the 2010 Air Pollutant Emission Inventory (APEI) and the United States 2011 National Emissions Inventory (NEI) for the anthropogenic Hg emissions data. We thank Christine Wiedinmyer of the National Center for Atmospheric Research for providing the FINN emissions. We thank Junhua Zhang and Michael Moran for the preparation of model-ready gridded Hg emissions used in this study. Finally, we thank Paul Makar for coordinating/managing the Oil Sands modelling studies and the Forest Fire Working Group which motivated and contributed to this study through securing funding and

insightful discussions.

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



Figure 1: Location of all of the Canadian stations and the American stations where fire plumes were observed. Filled symbols indicate stations at which a fire plume was observed, and these stations are labelled with the site names given in Table 1. Colours indicate the Canadian provinces and territories. Provincial abbreviations are as follows: BC = British Columbia, AB = Alberta, SK = Saskatchewan, MB = Manitoba, ON = Ontario, QC = Quebec, NB = New Brunswick, NS = Nova Scotia, PE =
Prince Edward Island, YT = Yukon Territory, NT = Northwest Territories, NU = Nunavut.

Figure 2: Daily GEM concentrations from observations and model with prior and posterior emission factors for all 30 fire plumes. The model concentrations have been corrected for the bias between the model and the observations. The shaded area indicates the time period designated as the fire plume event. Note the different y-scales.

Figure 2: Daily GEM concentrations from observations and model with prior and posterior emission factors for all 30 fire
plumes. The model concentrations have been corrected for the bias between the model and the observations. The shaded area indicates the time period designated as the fire plume event. Note the different y-scales.

Figure 3: Examples of fire plumes (shown as surface air concentrations of $Hg^0$) showing typical source regions that impact the labelled station. Dots denote the location of an impacted observation station from Table 1.

Figure 4: Daily fractional GEM concentration contributions of the six vegetation types to the 35 fire events. Values closer to one
(dark purple) indicate that more of the fires were in that vegetation type. The horizontal dotted lines separate the 35 fire events; wider gaps indicate fires that took place over a longer time frame.

Figure 5: Averaging kernels for the emission factors of the six vegetation types included in the optimization. The dotted line shows the area under the averaging kernel for each vegetation type.

Figure 6: Prior (a) and posterior (b) mercury emissions (t/year) from biomass burning by province/territory for the burning
season (May – September) for 2010-2015, the mean of the six-year time period, and the anthropogenic emissions for the burning season. For biomass burning, only GEM is released, for anthropogenic emissions TOM species are also released. (c) As (a) but relative contribution. (d) As (b) but relative contribution. Provincial and territorial name abbreviations are given in the caption of Figure 1. Because of their small contribution to the total, we group the Atlantic provinces (NB, NS, PE, and NL) together in one group, AT.

Figure 7: Mercury emissions from biomass burning in Canada for 2010-2015, given in $\mu g/m^2 \cdot month$, averaged over the burning season (May-September). Point source emissions were aggregated to 60 km grid.

Figure 8: Total mercury emissions (TAM) from anthropogenic activity in Canada, given in $\mu g/m^2 \cdot month$, averaged over the burning season (May-September).

Figure 9: Percent contribution to total surface GEM concentration of Canadian biomass burning and anthropogenic emissions for
burning seasons (May – September) of 2010-2015 for each province/territory. We have grouped the Atlantic provinces together as AT. Note the different y-scales. Inset numbers are the mean percentage contribution to the total concentration over the six-years.

Figure 10: Same as Figure 9, but for total surface TOM (GOM + PBM) concentration.

Figure 11: Same as Figure 9, but for the Canadian observation sites.

Figure 12: Same as Figure 10, but for the Canadian observation sites.

Figure 13: Total atmospheric mercury deposition (t/year) by province/territory for the burning seasons (May-September) of 2010-2015 from (a) prior, (b) posterior and (c) prior emissions with PBM mercury emissions from Canadian biomass burning. Also shown are the mean of the six-year time period and the mean of the deposition from Canadian anthropogenic emissions for the burning season. (d) Biomass burning and anthropogenic deposition contributions relative to the total Hg deposition in
respective provinces/territories during biomass burning season for prior with PBM biomass burning emissions scenario. Provincial/territorial abbreviations are given in the caption of Figure 1. As in Figure 6, we have grouped the Atlantic provinces into one group, AT.

Figure 14: Average total atmospheric mercury deposition from biomass burning (prior with PBM biomass burning emissions scenario) in Canada during the burning season (May – September) for 2010-2015, given in $\mu g/m^2 \cdot month$.

Figure 15: Percentage contribution of deposition from Canadian biomass burning emissions (prior with PBM biomass burning emissions scenario) to the total deposition from all sources during the burning season (May – September) for 2010-2015.

Figure 16: Total atmospheric mercury deposition from anthropogenic emissions in Canada during the burning season (May – September) for 2012, given in $\mu g/m2 \cdot month$.







**Figure 1: Location of all of the Canadian stations and the American stations where fire plumes were observed. Filled symbols indicate stations at which a fire plume was observed, and these stations are labelled with the site names given in Table 1. Colours indicate the**
5 **Canadian provinces and territories. Provincial abbreviations are as follows: BC = British Columbia, AB = Alberta, SK = Saskatchewan, MB = Manitoba, ON = Ontario, QC = Quebec, NB = New Brunswick, NS = Nova Scotia, PE = Prince Edward Island, YT = Yukon Territory, NT = Northwest Territories, NU = Nunavut.**





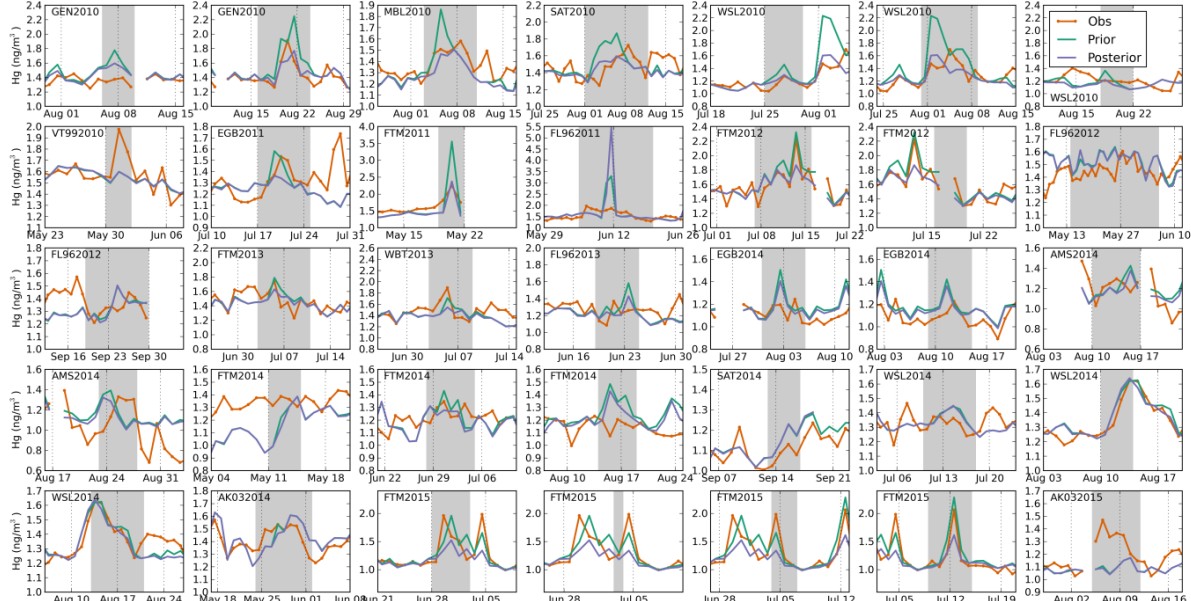

**Figure 2: Daily GEM concentrations from observations and model with prior and posterior emission factors for all 30 fire plumes. The model concentrations have been corrected for the bias between the model and the observations. The shaded area indicates the time period designated as the fire plume event. Note the different y-scales.**



**Figure 3: Examples of fire plumes (shown as surface air concentrations of Hg⁰) showing typical source regions that impact the labelled station. Dots denote the location of an impacted observation station from Table 1.**



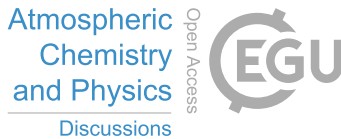

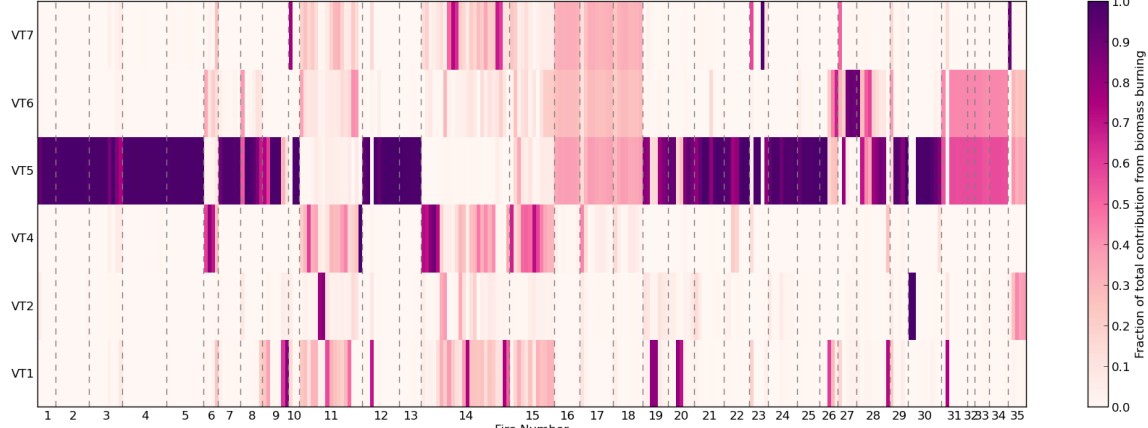

**Figure 4: Daily fractional GEM concentration contributions of the six vegetation types to the 35 fire events. Values closer to one (dark purple) indicate that more of the fires were in that vegetation type. The horizontal dotted lines separate the 35 fire events; wider gaps indicate fires that took place over a longer time frame.**





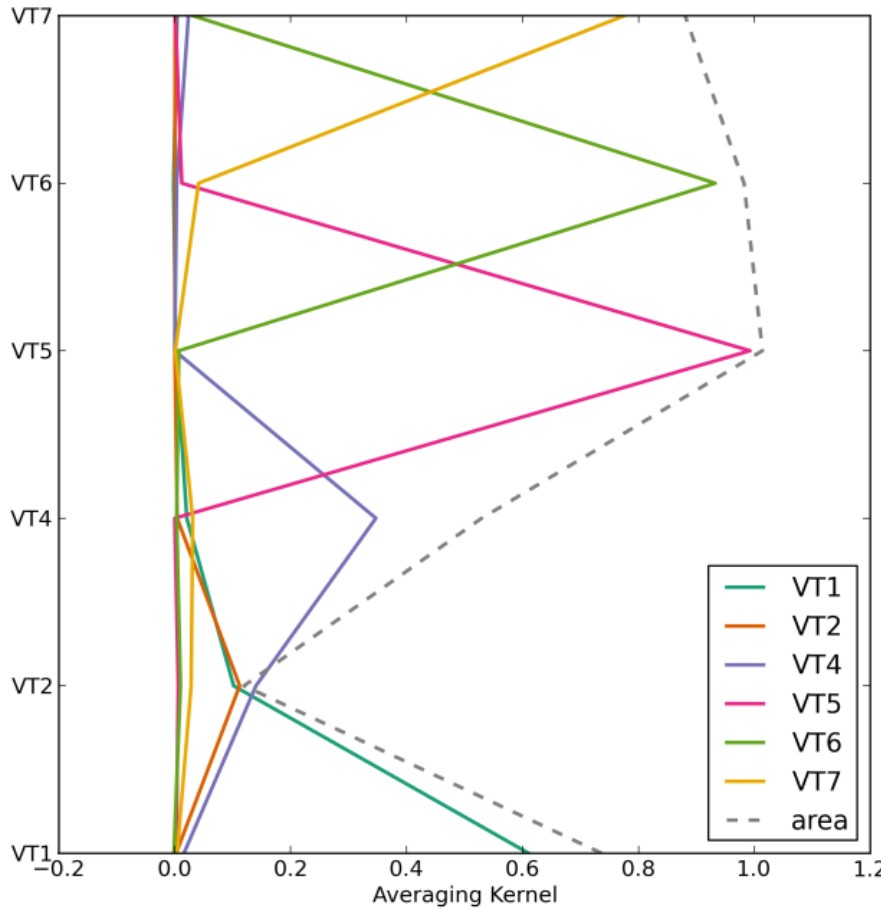

**Figure 5: Averaging kernels for the emission factors of the six vegetation types included in the optimization. The dotted line shows the area under the averaging kernel for each vegetation type.**



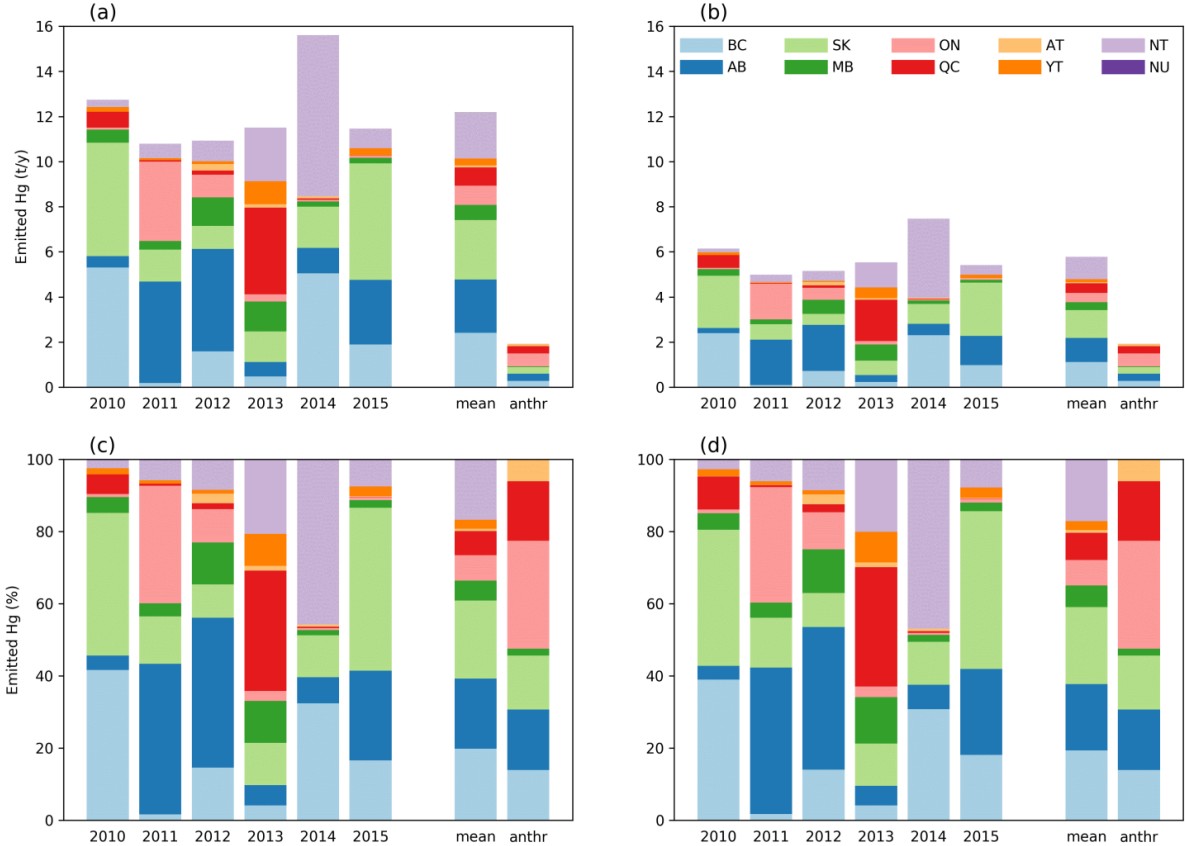

**Figure 6: Prior (a) and posterior (b) mercury emissions (t/year) from biomass burning by province/territory for the burning season
5 (May – September) for 2010-2015, the mean of the six-year time period, and the anthropogenic emissions for the burning season. For
biomass burning, only GEM is released, for anthropogenic emissions TOM species are also released. (c) As (a) but relative
contribution. (d) As (b) but relative contribution. Provincial and territorial name abbreviations are given in the caption of Figure 1.
Because of their small contribution to the total, we group the Atlantic provinces (NB, NS, PE, and NL) together in one group, AT.**

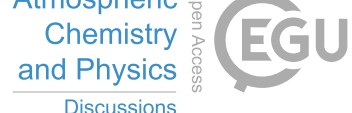



**Figure 7: Mercury emissions from biomass burning in Canada for 2010-2015, given in μg/m²·month, averaged over the burning season (May-September). Point source emissions were aggregated to 60 km grid.**





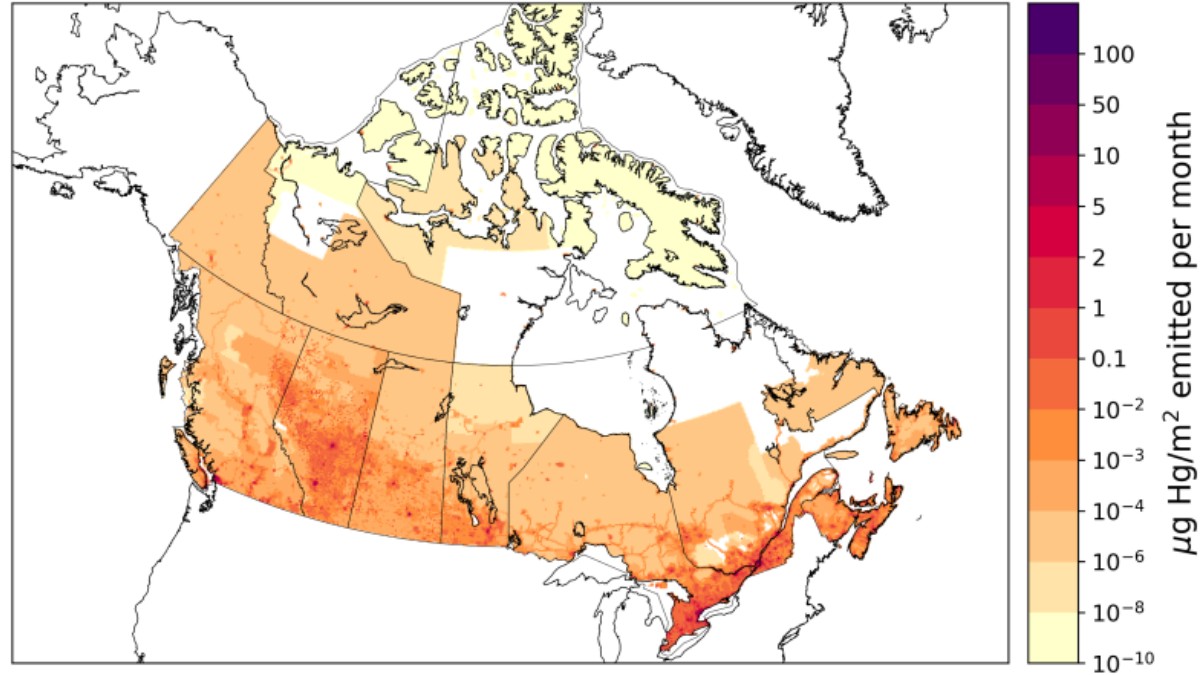

**Figure 8: Total mercury emissions (TAM) from anthropogenic activity in Canada, given in μg/m²·month, averaged over the burning season (May-September).**




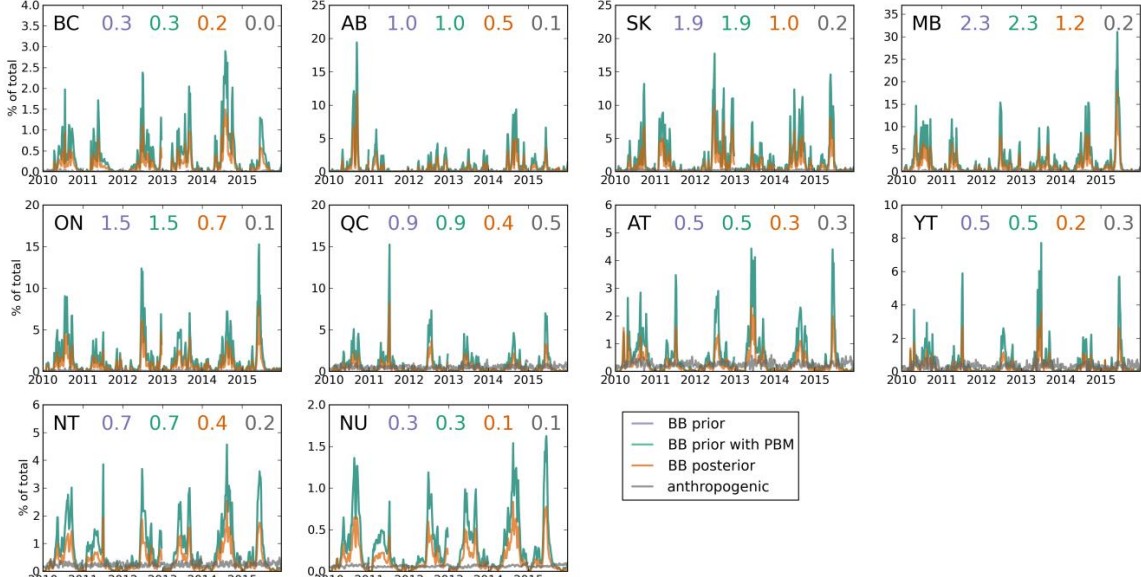

**Figure 9: Percent contribution to total surface GEM concentration of Canadian biomass burning and anthropogenic emissions for burning seasons (May – September) of 2010-2015 for each province/territory. We have grouped the Atlantic provinces together as AT. Note the different y-scales. Inset numbers are the mean percentage contribution to the total concentration over the six-years.**



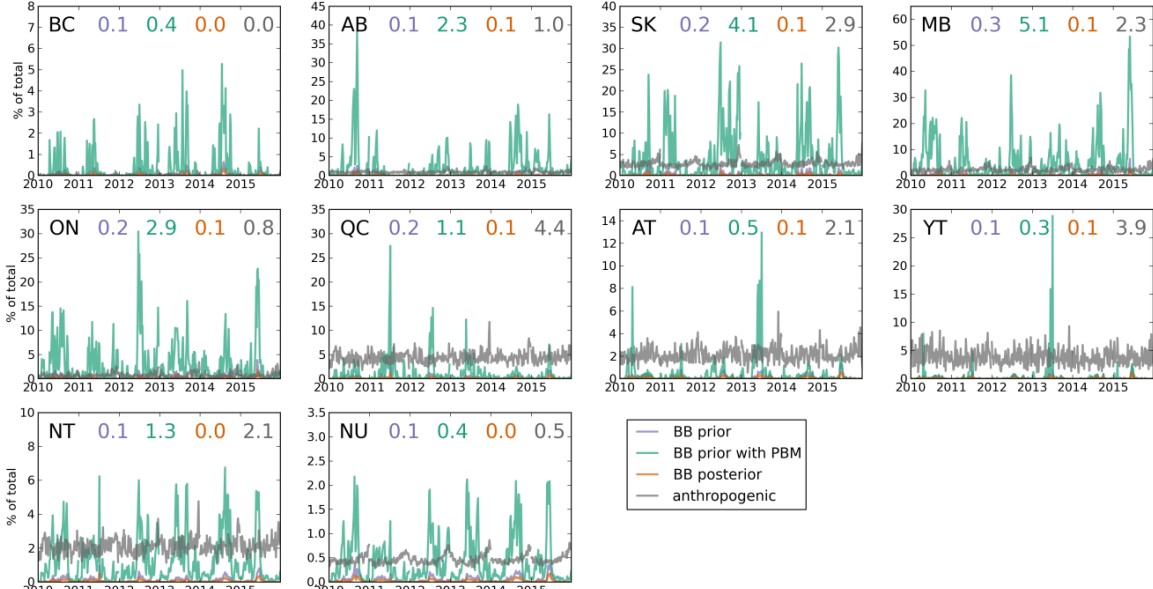

**Figure 10: Same as Figure 9, but for total surface TOM (GOM + PBM) concentration.**





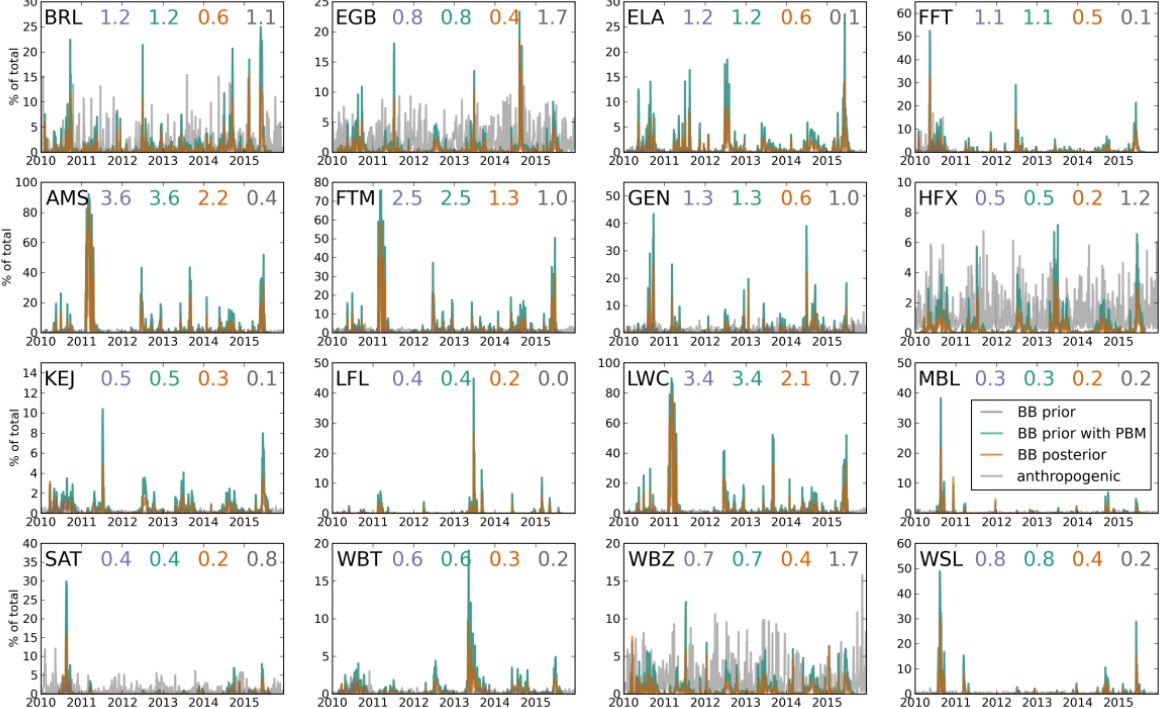

Figure 11: Same as Figure 9, but for the Canadian observation sites.





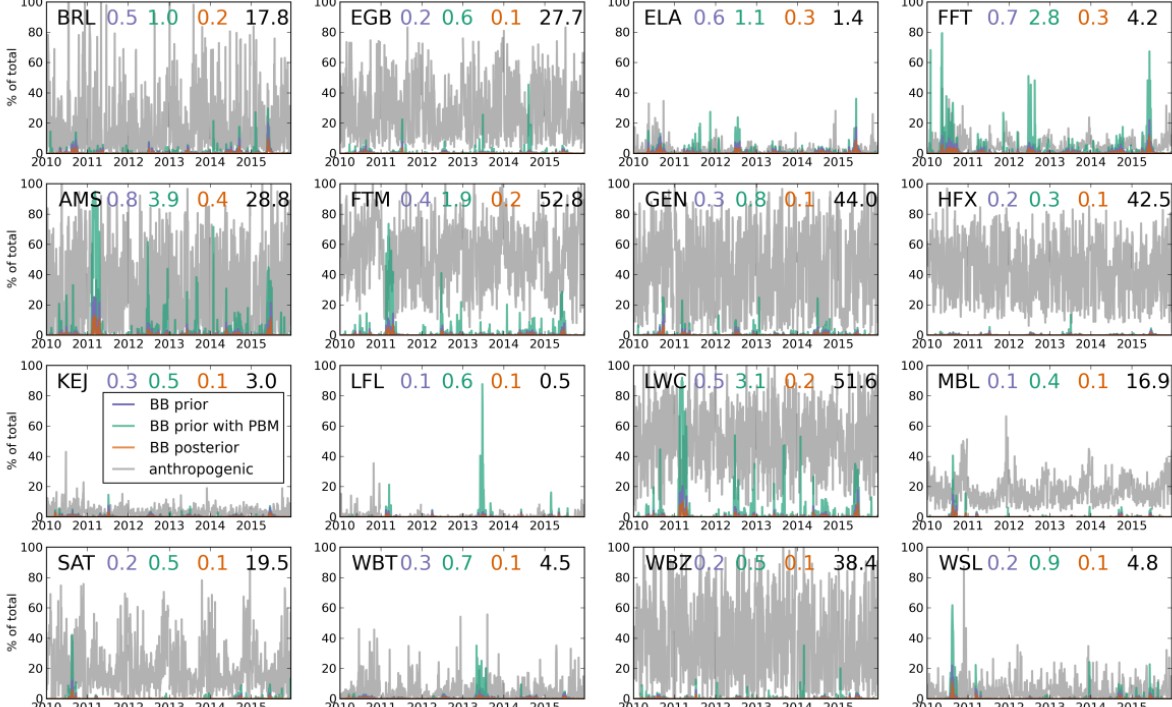

**Figure 12: Same as Figure 10, but for the Canadian observation sites.**



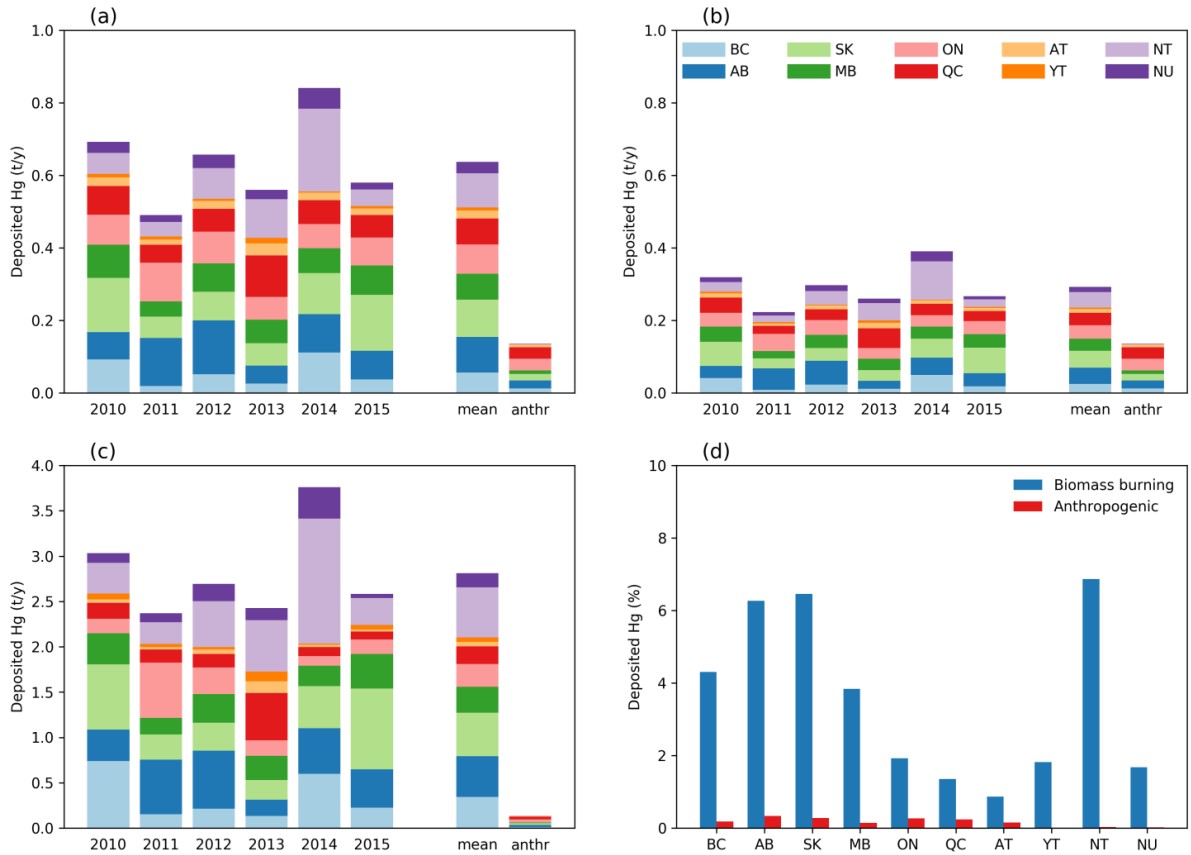

**Figure 13: Total atmospheric mercury deposition (t/year) by province/territory for the burning seasons (May-September) of 2010-2015 from (a) prior, (b) posterior and (c) prior emissions with PBM mercury emissions from Canadian biomass burning. Also shown are the mean of the six-year time period and the mean of the deposition from Canadian anthropogenic emissions for the burning season. (d) Biomass burning and anthropogenic deposition contributions relative to the total Hg deposition in respective provinces/territories during biomass burning season for prior with PBM biomass burning emissions scenario. Provincial/territorial abbreviations are given in the caption of Figure 1. As in Figure 6, we have grouped the Atlantic provinces into one group, AT.**



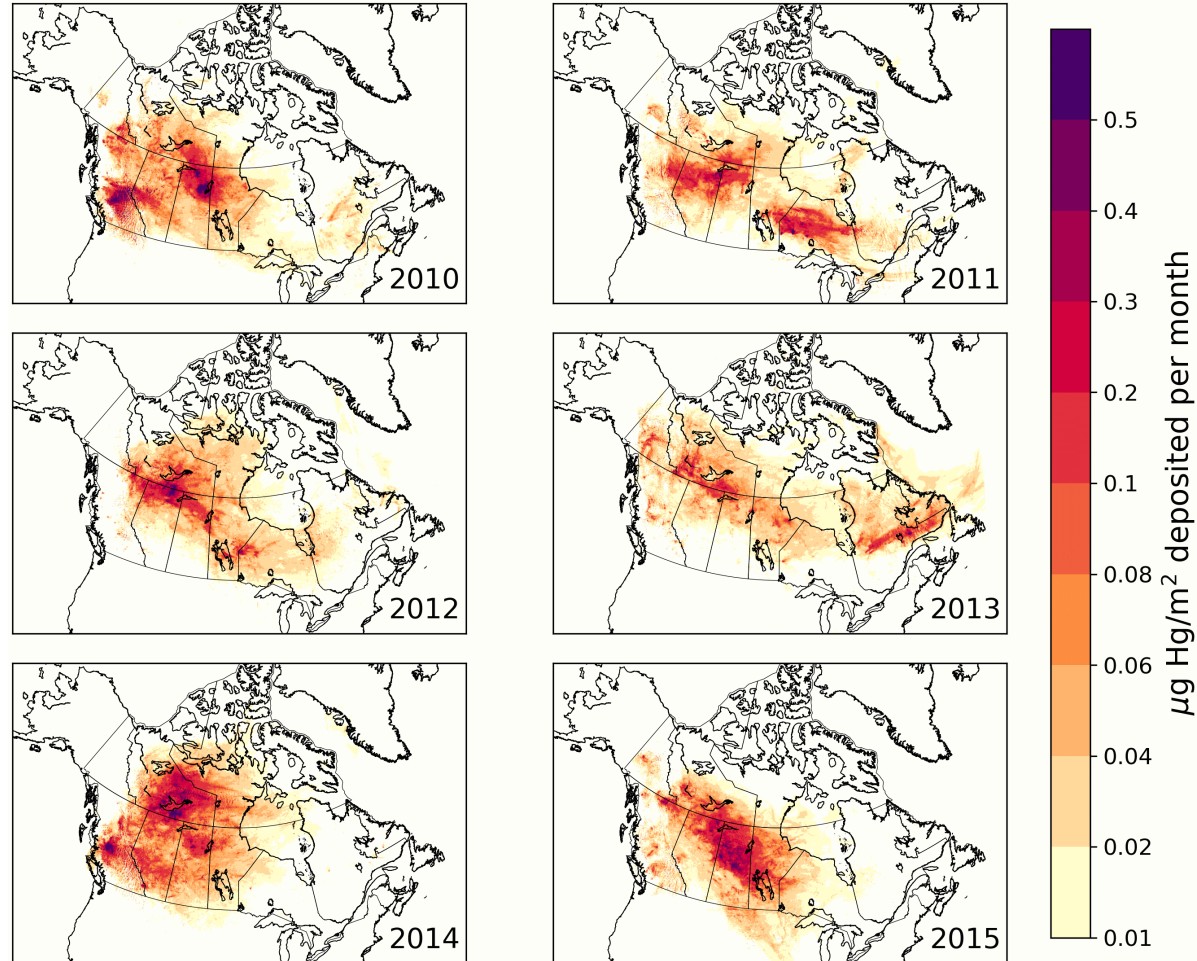

**Figure 14: Average total atmospheric mercury deposition from biomass burning (prior with PBM biomass burning emissions scenario) in Canada during the burning season (May – September) for 2010-2015, given in µg/m²·month.**



**Figure 15: Percentage contribution of deposition from Canadian biomass burning emissions (prior with PBM biomass burning emissions scenario) to the total deposition from all sources during the burning season (May – September) for 2010-2015.**



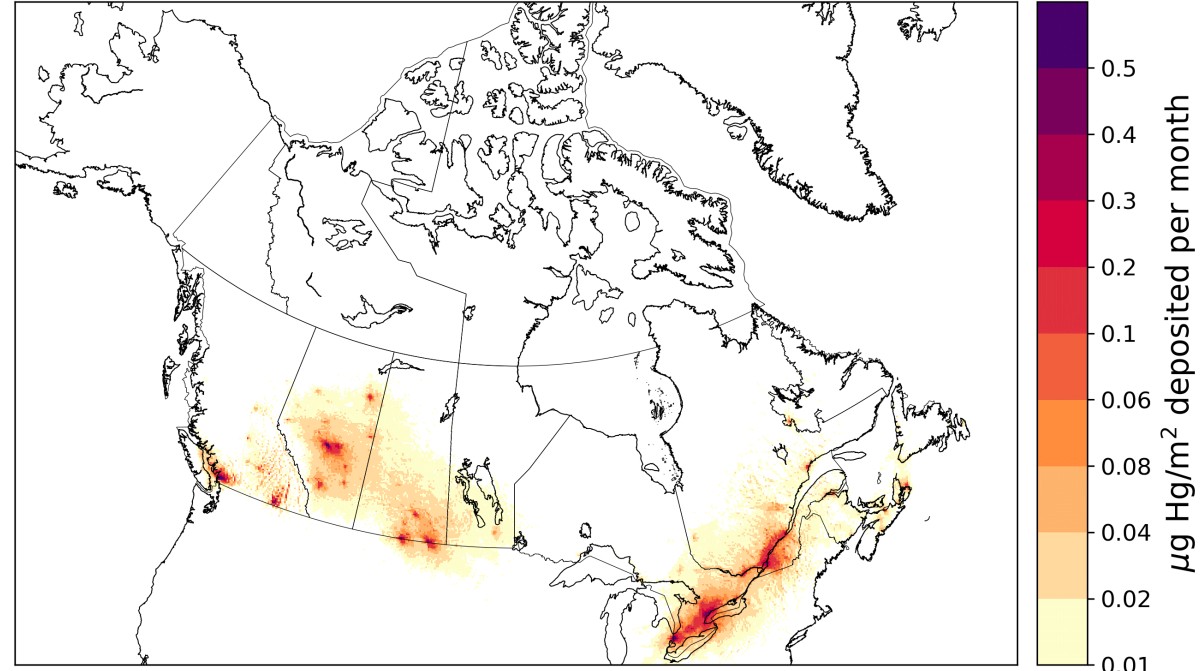

**Figure 16: Total atmospheric mercury deposition from anthropogenic emissions in Canada during the burning season (May – September) for 2012, given in μg/m²·month.**



**Table 1: Location of all Canadian observation stations and American stations where fire plumes were observed from 2010-2015. Full network names are given in the text. Sites that observed fires are indicated with an asterisk (*). End dates are the latest date that data was available; the station may still be operating.**

| Site Name | Location | Network | Latitude | Longitude | Ground altitude | Dates available |
|---|---|---|---|---|---|---|
| AMS* | Fort McKay South, AB | JOSM | 57.1 | -111.6 | 245 | 2014 |
| BRL | Bratt's Lake, SK | CAPMoN | 50.2 | -107.7 | 587 | 2001-2010 |
| EGB* | Egbert, ON | CAPMoN | 44.2 | -79.8 | 251 | 1996-2014 |
| ELA | Experimental Lakes Area, ON | Special Study | 49.6 | -93.7 | Not available | 2005-2010 |
| FTM* | Fort McMurray, AB | JOSM | 56.8 | -111.5 | 370 | 2010-2015 |
| FFT | Flin Flon, MB | Special Study | 54.7 | -101.8 | 335 | 2008-2015 |
| GEN* | Genesee, AB | Special Study | 53.3 | -114.2 | 807 | 2010 |
| HFX | Halifax, NS | Special Study | 44.6 | -63.5 | Not available | 2009-2010 |
| KEJ | Kejimkujik, NS | CAPMoN | 44.4 | -65.2 | 155 | 1995-2015 |
| LFL | Little Fox Lake, YT | NCP | 61.3 | -135.6 | 1128 | 2007-2014 |
| LWC* | Fort McMurray, AB | JOSM | 57.0 | -111.5 | 240 | 2012-2014 |
| MBL* | Ucluelet, BC | CAMNet | 48.9 | -125.5 | 15 | 2010-2014 |
| SAT* | Saturna, BC | CAPMoN | 48.7 | -123.1 | 178 | 2009-2014 |
| WBT* | Mingan, PQ | CAMNet | 50.3 | -64.2 | 11 | 1997-2014 |
| WBZ | St. Anicet, PQ | CAMNet | 45.2 | -74.0 | 49 | 1994-2014 |
| WSL* | Whistler, BC | CARA | 50.1 | -122.9 | 2182 | 2008-2014 |
| AK03* | Denali, AK | AMNet | 63.7 | -148.9 | 661 | 2014-2015 |
| FL96* | Pensacola, FL | AMNet | 30.5 | -87.3 | 45 | 2009-2015 |
| VT99* | Underhill, VT | AMNet | 44.5 | -72.8 | 399 | 2008-2015 |





**Table 2: Prior and posterior emission factors (EFs) used to generate emissions estimates. Prior EFs are from Wiedinmyer and Friedli (2007). All EFs are given in units of × 10⁻⁶ g Hg (kg fuel burned)⁻¹. The error reduction is given in brackets for the posterior emissions.**

| Vegetation type | Prior EF | Posterior EF | Posterior EF – no VT2 |
|---|---|---|---|
| VT1 – grassland | $274 \pm 274$ | $221 \pm 170$ (38%) | $213 \pm 170$ (38%) |
| VT2 – woody savanna | $41.1 \pm 41.1$ | $-29 \pm 38$ (7%) | *not optimized* |
| VT3 – tropical forest | 239 | *not optimized* | *not optimized* |
| VT4 – temperate forest | $239 \pm 239$ | $264 \pm 181$ (24%) | $254 \pm 181$ (24%) |
| VT5 – boreal forest | $315 \pm 315$ | $140 \pm 27$ (91%) | $140 \pm 27$ (91%) |
| VT6 – temperate needleleaf forest | $239 \pm 239$ | $315 \pm 62$ (74%) | $315 \pm 62$ (74%) |
| VT7 - crops | $274 \pm 274$ | $217 \pm 129$ (53%) | $215 \pm 129$ (53%) |



**Table 3: Pearson correlation coefficient (r), Root Mean Square Error (RMSE), Unbiased RMSE (UMRSE) and Normalized Standard Deviation (NSD) values between model simulated and observed surface air concentrations of GEM using all observation sites calculated for all individual years studied and collectively for all years using literature ("Prior") and optimized ("Posterior") biomass burning emission factors of Hg. The bolded values indicate better performance of the model.**

| Year | Prior | | | | Posterior | | | |
|------|------|------|-------|------|------|------|-------|------|
| | R | RMSE | URMSE | NSD | R | RMSE | URMSE | NSD |
| 2010 | 0.61 | 0.26 | 0.13 | 1.12 | **0.64** | **0.25** | **0.11** | **0.94** |
| 2011 | **0.53** | 0.26 | 0.24 | 1.65 | 0.49 | **0.25** | **0.22** | **1.55** |
| 2012 | **0.35** | **0.20** | 0.13 | **1.03** | 0.33 | 0.21 | **0.13** | 0.94 |
| 2013 | 0.50 | 0.21 | 0.16 | **0.91** | **0.55** | **0.20** | **0.15** | 0.80 |
| 2014 | 0.58 | 0.16 | 0.13 | **0.89** | **0.59** | **0.16** | **0.12** | 0.88 |
| 2015 | **0.76** | 0.18 | **0.13** | 1.19 | 0.72 | **0.17** | 0.13 | **1.07** |
| Avg | 0.56 | 0.21 | 0.15 | **1.10** | **0.57** | **0.20** | **0.14** | **1.00** |