# Peer review of "How important is biomass burning in Canada to mercury contamination?"

_Atmospheric Chemistry and Physics, 2017_

## Referee Comment (RC1) · Anonymous Referee #1 · 19 Oct 2017

This paper seeks to improve estimates of biomass burning emissions and investigates the impact of these emissions on Canadian mercury contamination by combining air quality modelling, observations from a network of atmospheric mercury concentration monitoring sites between 2010 and 2015, and Bayesian inversion techniques. The research questions are relevant to the ACP community, and this work uses appropriate methods to provide new and valuable insight into the importance of biomass burning as a source of emissions in Canada and the spatial distribution of its impact. I would recommend publication in ACP after minor revisions to: further clarify terminology, description of methods, and figures; and deepen the discussion to further highlight how this study contributes to our understanding of mercury from biomass burning more broadly.

[Figure]

1. "Biomass burning" as a term can be a little ambiguous, and in different communities is used to encompass some or all of: wildfires, crop residue burning, planned burning, and biofuel burning for heat and electricity generation. My impression is that in this study (given the data inputs), biomass burning is used to refer to the first three. It might be useful to spell this out explicitly early on in the paper to avoid any confusion.

2. (I've grouped a couple of related comments into this point — apologies if it is a little sprawling.) In the Abstract, the inversion is largely presented as a means to an ends, though in the results and discussion, the authors touch on several points that emerged from the inversion process that, to me, were also major contributions of the study that merited mention in the Abstract. Further elaboration on these points might be interesting to those studying mercury and biomass burning more generally. Some examples:

20-28 p.9: "This is an indication of a break down in one or more of our initial assumptions: the FINN calculation of burned biomass has uncertainties in magnitude or in location, the emission factors are not constant in space-time but are functions of fire type and other factors such as atmospheric deposition, or the six vegetation types do not accurately represent the variation in mercury emissions by species... Peat is much more 25 prevalent in the Northwest Territories than in BC (Tarnocai et al., 2011), the discrepancy in improvement between the years is perhaps an indication that peatland should be considered in defining the vegetation types; this is currently difficult due to sparseness of measurements of Hg from biomass burning plumes."

6-11, p.15: "Our synthesis inversion study could be improved upon by implementing a more detailed optimization scheme, for example by considering more vegetation/land-use types such as peatland into consideration when assigning vegetation types and by accounting for spatial distribution of atmospheric deposition. Comprehensive measurements of mercury species in biomass burning emission plumes for different land-use types, and a suitable network of air concentration measurements of mercury including speciation would help in constraining the estimates of the Hg emissions from biomass

burning and the resulting deposition."

Related to the above, could the authors elaborate a little further on this point of what we would need to better constrain biomass burning emissions using top down measurements? It certainly seems like GEM:PBM emissions ratio would be a useful parameter to conduct a synthesis inversion on, but that the speciated measurements are lacking. Could wet deposition measurements be helpful in this regard or are they unable to capture evidence of the plumes?

A brief summary of how factors like fire type, temperature etc. are thought to affect biomass burning emissions and how the different emissions inventories treat these issues might be helpful earlier in the manuscript as well. Is there a non-FINN alternative that better captures the factors that are relevant for North American biomass burning emissions?

3. L20, p.1: The phrase "the range" is used here and elsewhere in the manuscript, which I think may be a little misleading, as it suggests to me "the full range" and I'm not sure that the scenarios considered do represent the full range of potential emissions. As the authors note, there are other sources of uncertainty not considered, and also structural uncertainties in emissions calculation methods between inventories (e.g., FINN, GFED). Using the phrase "a range" might be more accurate.

4. L17-20, p.2: It might be worthwhile to flag here, or in the following sentence, that emissions from soils and oceans can be secondary emissions of originally anthropogenically emitted mercury.

5. L17, p.4: Should there be another study listed here, in addition to the Cole et al. 2014?

6. Section 4: I found it somewhat difficult to understand the fire events ID process, especially when looking at Figure 2. Is the same criteria used to determine whether there's a peak in the observed data?

7. L25-35, p.8: I found this discussion of the range of reported estimates of EFs (from which the priors were derived) quite helpful. I wonder whether some of this info could be included in Table 2?

8. Figure 1 caption: It reads "Location of all of the Canadian stations and the American stations where fire plumes were observed. Filled symbols indicate stations at which a fire plume was observed, and these stations are labelled with the site names given in Table 1." What do the open circles mean?

9. Figure 2 caption (and others): "The model concentrations have been corrected for the bias between the model and the observations." Was this bias correction discussed in the manuscript?

10. Figure 4 caption: It might be helpful for the reader to use the full names of the vegetation types here rather than the codes, either in the figure itself or to list them in the caption.

11. Figure 7 caption: This is mentioned in the text, but it would be useful to include in the caption also which of the three emissions scenarios this is for.

---

## Referee Comment (RC2) · Anonymous Referee #2 · 22 Nov 2017

The presented paper investigates the impact of wild fires on atmospheric transport of mercury and Hg wet deposition in Canada (and North America). The impact of natural (re-)emissions of Hg on global and regional transport of Hg is an important research question. The employed state-of-the-art CTM and the used methods are appropriate. Moreover, the manuscript is well written. Thus I can recommend publication in ACP after a few (mostly minor) revisions:

1) I am missing a discussion on the impact of the general model bias on the EF optimization. In my opinion it would be necessary to first investigate the model performance at the measurement site at times without fire events.

2) Please clarify: Did you use the complete 6 year run to calculate the posteriori EFs?

[Figure]

3) As I understand one main result is that the PBM fraction of Hg emitted from wildfires has a much larger influence on modelled Hg wet deposition than the uncertainty in the emission factors. In this regard, I think that the brief discussion (compared to the discussion of EFs) of the PBM fraction is not adequate and needs to be improved.

First of all I miss a detailed analysis of the impact of different PBM fractions in the wild fire emissions on model performance. Page 10 lines 10-20: Here you must include a table showing statistics on the impact of increased PBM emissions on model-observation comparisons. (similar to Table 3)

4) Table 2: VT-2 becomes a sink for mercury? If you choose to show this it needs to be discussed in the text.

5) In table 3 please indicate for which years the difference between the sensitivity runs is statistically significant.

6) Figure 13 d) Please include all runs (a,b,c) in this figure.

7) The manuscript includes numerous double spaces that need to be removed.

---

## Author Comment (AC1) · 29 Mar 2018

We would like to thank both reviewers for their careful review of our manuscript. Their comments have helped to improve the manuscript. In what follows we repeat the original comments in italic, followed by our responses.

**Reviewer 1:**

*This paper seeks to improve estimates of biomass burning emissions and investigates the impact of these emissions on Canadian mercury contamination by combining air quality modelling, observations from a network of atmospheric mercury concentration monitoring sites between 2010 and 2015, and Bayesian inversion techniques. The research questions are relevant to the ACP community, and this work uses appropriate*

[Figure]

*methods to provide new and valuable insight into the importance of biomass burning as a source of emissions in Canada and the spatial distribution of its impact. I would recommend publication in ACP after minor revisions to: further clarify terminology, description of methods, and figures; and deepen the discussion to further highlight how this study contributes to our understanding of mercury from biomass burning more broadly.*

*1. "Biomass burning" as a term can be a little ambiguous, and in different communities is used to encompass some or all of: wildfires, crop residue burning, planned burning, and biofuel burning for heat and electricity generation. My impression is that in this study (given the data inputs), biomass burning is used to refer to the first three. It might be useful to spell this out explicitly early on in the paper to avoid any confusion.*

The following sentence has been added at the end of the second paragraph in the introduction to clarify the sources of biomass burning in our study:

In our study, biomass burning includes wildfire, agricultural fires, and prescribed burning.

*2. (I've grouped a couple of related comments into this point — apologies if it is a little sprawling.) In the Abstract, the inversion is largely presented as a means to an ends, though in the results and discussion, the authors touch on several points that emerged from the inversion process that, to me, were also major contributions of the study that merited mention in the Abstract. Further elaboration on these points might be interesting to those studying mercury and biomass burning more generally. Some examples: 20-28 p.9: "This is an indication of a break down in one or more of our initial assumptions: the FINN calculation of burned biomass has uncertainties in magnitude or in location, the emission factors are not constant in space-time but are functions of fire type and other factors such as atmospheric deposition, or the six vegetation types do not accurately represent the variation in mercury emissions by species. . . Peat is much more 25 prevalent in the Northwest Territories than in BC (Tarnocai et al.,*

*2011), the discrepancy in improvement between the years is perhaps an indication that peatland should be considered in defining the vegetation types; this is currently difficult due to sparseness of measurements of Hg from biomass burning plumes."*

*6-11, p.15: "Our synthesis inversion study could be improved upon by implementing a more detailed optimization scheme, for example by considering more vegetation/landuse types such as peatland into consideration when assigning vegetation types and by accounting for spatial distribution of atmospheric deposition. Comprehensive measurements of mercury species in biomass burning emission plumes for different land-use types, and a suitable network of air concentration measurements of mercury including speciation would help in constraining the estimates of the Hg emissions from biomass burning and the resulting deposition."*

We have added the following to the abstract:

The inversion results suggest that EFs representing more vegetation types – specifically peatland – are required. This is currently limited by the sparseness of measurements of Hg from biomass burning plumes. More measurements of Hg concentration in the air, specifically downwind of fires, would also improve the inversions.

*Related to the above, could the authors elaborate a little further on this point of what we would need to better constrain biomass burning emissions using top down measurements? It certainly seems like GEM:PBM emissions ratio would be a useful parameter to conduct a synthesis inversion on, but that the speciated measurements are lacking.*

With only 30 fire events observed during a six-year dataset, the current inversion system is primarily hampered by a lack of observations. The observation network was not designed to observe biomass burning events, and so locations of sites are not ideal for this work. More observation sites placed strategically to capture biomass burning events in more vegetation types would increase the number of fires and improve the inversions.

As we have shown, the speciation of mercury from biomass burning is highly uncertain and has significant consequences for transport away from the fire. Observations of speciated mercury would be useful for validating the model output, and could also be incorporated into an improved inversion system to constrain the speciation ratios.

We have expanded the pre-existing discussion of this in the manuscript, pg. 15, line 9:

...of atmospheric deposition. With only 30 fire events observed in a six-year dataset, with most of them occuring in boreal forest, the inversion system presented here is primarily hampered by a lack of observations. Comprehensive measurements of mercury species in biomass burning emission plumes for different land-use types, and a suitable network of air concentration measurements of mercury would be beneficial. Observations of speciated mercury would also be invaluable to help in constraining the estimates of the Hg emissions from biomass burning and the resulting deposition.

*Could wet deposition measurements be helpful in this regard or are they unable to capture evidence of the plumes?*

Wet deposition observations have coarse temporal resolution, usually two weeks accumulation period, compared to ambient measurements of GEM concentrations (i.e., minutes to daily). Wet deposition measurements cannot be used to catch biomass burning events, which typically last on the order of a couple days.

*A brief summary of how factors like fire type, temperature etc. are thought to affect biomass burning emissions and how the different emissions inventories treat these issues might be helpful earlier in the manuscript as well.*

The following text has been added in the revised manuscript (as Section 1.1 – Biomass burning inventories and their uncertainties) discussing different biomass burning emission inventories and their uncertainties.

[revised manuscript text omitted]

*Is there a non-FINN alternative that better captures the factors that are relevant for North American biomass burning emissions?*

We also compared the data to the model run using GFAS (Kaiser et al., 2012) and FEER (Ichoku and Ellison, 2014) inventories. (These comparisons are not in the paper.) We found that GEM-MACH-Hg run with FINN inventories was best able to reproduce the observed data during the fire events, and so we chose to do the experiment using FINN.

*3. L20, p.1: The phrase "the range" is used here and elsewhere in the manuscript, which I think may be a little misleading, as it suggests to me "the full range" and I'm not sure that the scenarios considered do represent the full range of potential emissions. As the authors note, there are other sources of uncertainty not considered, and also structural uncertainties in emissions calculation methods between inventories (e.g., FINN, GFED). Using the phrase "a range" might be more accurate.*

The manuscript has been revised by replacing the phrase "the range" to "a range" as suggested by the reviewer.

*4. L17-20, p.2: It might be worthwhile to flag here, or in the following sentence, that emissions from soils and oceans can be secondary emissions of originally anthropogenically emitted mercury.*

We have added the following sentence at the end of the paragraph in the revised manuscript as suggested by the reviewer:

It should be noted here that present day emissions from soils and oceans include revolatilization of originally anthropogenically emitted mercury.

*5. L17, p.4: Should there be another study listed here, in addition to the Cole et al. 2014?*

Cole et al. 2014 reports and provides an analysis of ambient measurement of mercury in Canada for all stations, and includes all relevant references. We have revised the citation to say "Cole et al. 2014 and references therein" in the revised manuscript.

*6. Section 4: I found it somewhat difficult to understand the fire events ID process, especially when looking at Figure 2. Is the same criteria used to determine whether there's a peak in the observed data?*

Yes, the same definition of a peak is applied to the model output and observations. While the peak determination is done on the difference between the model run with and without biomass burning (and the observations with the mean model with no biomass burning subtracted), Figure 2 shows the model run with biomass burning and the observations only. As a result the observed fire events are not always in the same place as the peaks. We have reworded the relevant paragraph to hopefully make this section more clear:

To identify fire plumes in the modeled and observed time series of GEM air concentrations at the observation sites, we run the model once with the complete global Hg emissions as described in Sect. 2, and once with all of the emissions except biomass burning Hg emissions in North America (i.e., the "no fire" run). The difference of these model runs gives us the GEM concentration as a result of only the biomass burning emissions in North America (i.e. the "fire only" model). We sample this difference in GEM concentration at the time and location of the observations. GEM peaks in these

station time series indicate times when the model predicts a fire plume at one of the stations. We compare the model simulated fire only GEM concentration to the observations with the mean of the "no fire" simulation GEM concentration subtracted. We define a fire event as any time the fire only model and observations have peaks that are within a day of one another, with a maximum value of the modelled GEM concentration greater than twice the standard deviation of the "no fire" modelled GEM concentration for that year and station. We apply the same definition of a peak to the observed data. The GEM peaks that are predicted by the model but not found in the observed data are attributed to model transport error or errors in the FINN burned biomass inventory and not errors in the emission factors, and these fire events are not considered in our analysis. Also, GEM peaks that are in the observations but not in the model output we assume to be from sources other than biomass burning. Using the model we follow the plume back in time to identify the source region of the fire. In total, we find 30 fire plumes in our six year dataset totalling 268 burning days, which are shown in Figure 2. Note that what is plotted in this figure are the uncorrected observations and model output, (i.e. no subtraction of "no fire" run), and so some of the peaks are due to other sources than biomass burning.

*7. L25-35, p.8: I found this discussion of the range of reported estimates of EFs (from which the priors were derived) quite helpful. I wonder whether some of this info could be included in Table 2?*

The values of EFs from various field studies are fully reported in Wiedinmyer and Friedli (2007) and we have referenced this paper. In our opinion, the range of these values as presented in the text is sufficient.

*8. Figure 1 caption: It reads "Location of all of the Canadian stations and the American stations where fire plumes were observed. Filled symbols indicate stations at which a fire plume was observed, and these stations are labelled with the site names given in Table 1." What do the open circles mean?*

Open circles represent stations where no plumes were observed. This is clarified in the caption:

Figure 1: Location of all of the Canadian stations and only the American stations where fire plumes were observed. Filled symbols indicate stations at which a fire plume was observed, and these stations are labelled with the site names given in Table 1. Open symbols indicate Canadian stations where no plumes were observed during our study.

*9. Figure 2 caption (and others): "The model concentrations have been corrected for the bias between the model and the observations." Was this bias correction discussed in the manuscript?*

The reviewer is correct that this bias correction was not discussed in the manuscript. We have added a discussion on page 7, line 18:

...shown in Figure 2. We corrected the model for bias from the observations using the average difference between the model and data during non-fire events. The largest observed bias was 0.4 ng/m3, though most were less than 0.1 ng/m3 . In the case of ...

*10. Figure 4 caption: It might be helpful for the reader to use the full names of the vegetation types here rather than the codes, either in the figure itself or to list them in the caption.*

Full names of the vegetation types have been added in the caption.

*11. Figure 7 caption: This is mentioned in the text, but it would be useful to include in the caption also which of the three emissions scenarios this is for.*

The figure caption has been revised to include the name of the emission scenario.

**Reviewer 2:**

*The presented paper investigates the impact of wild fires on atmospheric transport of mercury and Hg wet deposition in Canada (and North America). The impact of natural*

*(re-)emissions of Hg on global and regional transport of Hg is an important research question. The employed state-of-the-art CTM and the used methods are appropriate. Moreover, the manuscript is well written. Thus I can recommend publication in ACP after a few (mostly minor) revisions:*

*1) I am missing a discussion on the impact of the general model bias on the EF optimization. In my opinion it would be necessary to first investigate the model performance at the measurement site at times without fire events.*

Extensive verification of the model used in this study at comprehensive observation sites is available in previous publications of the model, already cited in the manuscript. Table 3 provides averaged verification of the model at the measurement sites for the complete period of model simulations for each year (May to September) including fire events. Since there are a significant number of small fire events local to the sites which contribute to GEM concentrations at the sites during the model simulation period, it is not possible to completely isolate the periods of fire events from observed time series for a comparison with model simulated values without fires. Also, background GEM concentrations at the sites are affected by regional fire events. Verification of model simulations including all emissions as performed in the table 3 is appropriate in this study.

We performed a series of Observation System Simulation Experiments (OSSEs) using simulated observations with known error and bias. Including a bias of 0.4 ng/m3 (the maximum observed bias) distributed randomly over the observation sites changes the retrieved EFs by less than 15

*2) Please clarify: Did you use the complete 6 year run to calculate the posteriori EFs?*

We only use the data during the identified fire events. We have added this point to the manuscript (page 8, line 2): …in Table 2. The optimization was performed using the daily-averaged data and model output for the days identified as having a fire event (Section 4). Since we mainly…

[Figure]

*3) As I understand one main result is that the PBM fraction of Hg emitted from wildfires has a much larger influence on modelled Hg wet deposition than the uncertainty in the emission factors. In this regard, I think that the brief discussion (compared to the discussion of EFs) of the PBM fraction is not adequate and needs to be improved.*

We do not state, "the PBM fraction of Hg emitted from wildfires has a much larger influence on modelled Hg wet deposition than the uncertainty in the emission factors". Separate emission factors for all Hg species are currently not available and, based on field observations, biomass burning emissions of Hg are believed to be in the form of GEM. Based on limited laboratory study results, we tested an additional biomass burning emission scenario by adding biomass burning emissions of PBM and show that this has a significant impact on total deposition (both dry and wet deposition). Since PBM emissions from biomass burning are not studied well to determine its fraction in total biomass burning mercury emissions, it is not possible to add further discussion.

*First of all I miss a detailed analysis of the impact of different PBM fractions in the wild fire emissions on model performance. Page 10 lines 10-20: Here you must include a table showing statistics on the impact of increased PBM emissions on model-observation comparisons. (similar to Table 3)*

Figures 2 and Table 3 discuss the impacts of various biomass burning emission scenarios on model simulated GEM concentrations in comparison with observed values. Since biomass burning emissions of GEM are unchanged in the emission scenario with increased PBM emissions (same as the prior emission scenario; see paragraph 4 in this section), the impact of this emission scenario on model simulated GEM concentrations is the same as the prior emission scenario; therefore no additional information can be added to figure 2 or table 3 pertaining to increased PBM emission scenario. Additionally, measurements of air concentrations of PBM are not available for a model-measurement comparison as performed for GEM in figure 2 and table 3. Impacts of increased PBM emission scenario on air concentrations of mercury species and deposition are already discussed in Section 7: Biomass burning impacts on mercury burden

in Canada.

*4) Table 2: VT-2 becomes a sink for mercury? If you choose to show this it needs to be discussed in the text.*

We do not interpret this as a sink in mercury, but rather a poorly-constrained variable in the inversion, given the averaging kernels and the error reduction. We have added a note to the manuscript discussing this (page 8, line 20):

. . .contain one of more of the others. We interpret the negative value of VT2 shown in Table 2 to be a reflection of a poorly constrained variable in the optimization, and not that VT2 becomes a sink for mercury. Because of this analysis. . .

*5) In table 3 please indicate for which years the difference between the sensitivity runs is statistically significant.*

We have indicated which runs are statistically significant in the revised manuscript.

*6) Figure 13 d) Please include all runs (a,b,c) in this figure.*

Figure 13 d) has been revised as suggested by the reviewer (See Fig. 1 at the end of this response).

*7) The manuscript includes numerous double spaces that need to be removed.*

This formatting issue that will be resolved when the manuscript is typeset for publication.

[Figure]

**Fig. 1.**